# Towards Cognitively-Faithful Decision-Making Models to Improve AI Alignment

**Cyrus Cousins**[*]
Duke University

**Vijay Keswani**
IIT Delhi

**Vincent Conitzer**
CMU

**Hoda Heidari**
CMU

**Jana Schaich Borg**
Duke University

**Walter Sinnott-Armstrong**
Duke University

## ABSTRACT

Recent AI trends seek to align AI models to learned *human-centric objectives*, such as personal preferences, utility, or societal values. Using standard preference elicitation methods, researchers and practitioners build models of human decisions and judgments, to which AI models are aligned. However, standard elicitation methods often fail to capture the cognitive processes behind human decision making, such as *heuristics* or simplifying *structured thought patterns*. To address this failure, we take an axiomatic approach to learning *cognitively faithful* decision processes from pairwise comparisons. Building on the literature analyzing cognitive processes that shape human decision-making, we derive a model class in which features are first *processed* with learned rules, then *aggregated* via a fixed rule, such as the Bradley-Terry rule, to produce a decision. This structured processing of information ensures that such models are realistic and feasible candidates to represent underlying human decision-making processes. We demonstrate the efficacy of this modeling approach by learning interpretable models of human decision making in a kidney allocation task, and show that our proposed models match or surpass the accuracy of prior models of human pairwise decision-making.

## 1 INTRODUCTION

Computational models of human cognition allow us to quantify and study the factors that influence our decisions, and when accurate, they help explain commonalities in human decision processes across different tasks (Gigerenzer et al., 2000; Crockett, 2016). This may also be why computational models to predict human behavior have found applications in personalized AI tools in various settings, e.g., healthcare, autonomous vehicles, and resource allocation, to increase users' confidence that AI will be aligned with their preferences (Ji et al., 2023; Kim et al., 2018; Noothigattu et al., 2019).

Despite promising use cases, current AI models of human decision-making typically do not attempt to faithfully replicate cognitive processes. Modern AI tools built using preference elicitation (Lee et al., 2019; Awad et al., 2018; Johnston et al., 2023; Freedman et al., 2020; Rafailov et al., 2023) and reinforcement learning (Ng et al., 2000; Christiano et al., 2017; Kaufmann et al., 2023) implicitly assume that a reward/objective model based on a prespecified hypothesis class can accurately predict human decisions, but such tools are agnostic about how faithful these models are to cognitive processes. However, recent works highlight the lack of suitable hypothesis classes to capture human choices or their values (Stray et al., 2021; Boerstler et al., 2024; Leike et al., 2018). Unsuitable modeling classes introduce the possibility of inappropriate optimization formulations and *reward hacking* (Pan et al., 2022; Hubinger et al., 2019), and can lead to arbitrarily erroneous inferential models learned from human responses (Zhuang & Hadfield-Menell, 2020; Pan et al., 2022). Additionally, such misaligned models are limited in their explanatory power, undermining any *intrinsic trust* placed in the AI's decisions (Jacovi et al., 2021). These problems are further exacerbated when AI is used in high-stakes domains, including moral domains such as healthcare and sentencing (Sinnott-Armstrong et al., 2021; Kim et al., 2018), where stakeholders often expect an AI to justify its decision in a similar manner

---
[*]Correspondence to `originalcyruscousins@gmail.com`

and to the same extent as humans (Lima et al., 2021). Modeling done in prior moral decision-making contexts either favors simple model classes — e.g., linear models and decision trees (Lee et al., 2019; Noothigattu et al., 2018; Freedman et al., 2020), whose fit for human decision-making is highly context-specific (Ganzach, 2001; Zorman et al., 1997), or employs uninterpretable classes — e.g., neural networks or random forests (Wiedeman et al., 2020; Ueshima & Takikawa, 2024) — whose decision processes cannot be validated due to their uninterpretable designs. A qualitative study by Keswani et al. (2025) note this *process misalignment* as a crucial issue around AI-trust in morally-salient tasks like kidney transplant allocation, with one study participant expressing concern: *"they don't think like a human... what they might think should be ranked as priority, I may not"*.

Such human concerns highlight that, for certain real-world applications, building trustworthy AI tools requires accurately simulating human decision processes. However, human decision processes are often not captured by standard hypothesis classes. For example, consider humans' use of decision *heuristics*, defined by Gigerenzer & Gaissmaier (2011) as a strategy *"that ignores part of the information, with the goal of making decisions more quickly, frugally, and/or accurately than more complex methods."* Keswani et al. (2025) document several heuristics people use when deciding which patient should get an available kidney. Their study participants often used a form of *hiatus heuristic*, employing *thresholds* to isolate relevant patient feature information; e.g., some treated "number of dependents" as a binary (none or any), disregarding the actual *quantity*. Similarly, rather than contrast patients holistically, some participants used simple aggregation methods, such as the *tallying heuristic*, which simply counts the number of features favoring each patient to make their choices. While the role of such heuristics in human decision processes is well established (Gigerenzer & Gaissmaier, 2011; Shah & Oppenheimer, 2008; Gigerenzer et al., 2000), models learned from human decisions using standard hypothesis classes often don't capture or explain the heuristics people use to make decisions. For example, if a decision-maker uses the threshold-based heuristic mentioned above, linear models would fail to capture it, while neural networks or multivariate monotonic regressors may learn the heuristic, but would fail to explain its role in the decision process, instead representing it as some approximately-equivalent but hopelessly opaque process. To explain or simulate human decision processes accurately, we need methods to learn and represent these heuristics (and other decision-making nuances). To that end, this paper explores hypothesis classes that more accurately capture the cognitive processes people use to make their decisions in pairwise comparison settings.

This work presents a learning framework for computational models of pairwise human decision-making, which is motivated by prior work on human cognition. Classical *theoretical* axiomatic models of choice impose strong structural assumptions on preferences, e.g., strong *transitivity* concepts and *independence of irrelevant alternatives* (von Neumann & Morgenstern, 1944; Luce et al., 1959). However, decades of *empirical work* demonstrate that these assumptions are systematically violated in human decision-making, particularly in comparative and context-dependent settings (Tversky, 1969; Tversky & Kahneman, 1974). We thus propose a set of *substantially weaker* axioms (section 3) that constrain how comparative judgments are made, and from these axioms, we derive a class of feasible *cognitively faithful* decision-making models. This allows us to retain interpretability and theoretical grounding without contradicting empirically observed heuristic decision processes. Indeed, because our axioms are weaker than classical alternatives, they do not *fully specify* a decision making process, but rather, they *constrain the feasible space* of decision making. The approach is pragmatic: Our axioms are somewhat prescriptive, as they represent normatively-desirable attributes of decision making, but they also have a descriptive element, as they relax existing classical axioms that were themselves attempts to describe human behavior. Similarly, the violation of our axioms is *not necessarily* a critical failure of human reasoning, but as relatively simple claims, the axioms can be understood more easily than complex models, and they can be studied qualitatively and assessed for their replicability in AI systems that align to human decisions.

Given any pairwise comparison between two options $x_1, x_2 \in \mathbb{R}^d$, the decision-maker's preference is characterized by a two-stage process (section 3.1). The first stage captures the *editing rules* that the decision-maker employs over individual features $x_1^{(j)}, x_2^{(j)}$ to process/simplify/transform the information presented in feature $j \in [d]$. Each feature $j$ may have its own decision rule, and these rules work towards transforming $x_i^{(j)}$ in a manner that reflects the feature's contribution to the decision-maker's final choice (e.g., whether the decision-maker thresholds, ignores, linearly transforms, or otherwise processes the feature). We allow this transformation to be *contextual* in nature, whereby the value of one or more features can influence the transformation used for another feature. Such *conditional transformations* capture feature interactions in our hypothesis

class, increasing the models' expressiveness while ensuring that we still learn feature-level decision rules. The second stage captures the decision rule that aggregates the processed features to choose the *preferred option*. This aggregation rule can be as simple as the *tallying heuristic* (Gigerenzer et al., 2000) or as complicated as *Bradley-Terry* (1952) *aggregation* for probabilistic preferences. Beyond the motivations from prior works in cognitive science and preference modeling, we show that our axiomatic basis yields the two-stage model class (section 3.2). Moreover, more stringent assumptions yield special cases of interest, such as logistic regression, probit regression, and monotonic decision rules. Finally, we assess the empirical efficacy of our proposed framework over synthetic and real-world datasets in the kidney allocation domain (section 4), a domain where users have said cognitive process alignment is particularly important. We find that our method can learn models that explain the decision rules that people use for kidney allocation decisions, while ensuring that the learned model has similar or better predictive accuracy than other modeling approaches.

## 2 RELATED WORK

Methods for learning preferences from comparative choices have been proposed in many contexts, such as for recommender systems (Kalloori et al., 2018; Guo & Sanner, 2010; Chen & Pu, 2004), language model personalization (Rafailov et al., 2023; Jiang et al., 2024; Ziegler et al., 2019; Ouyang et al., 2022), and explanatory frameworks in psychology (Lichtenstein & Slovic, 2006; Ben-Akiva et al., 2019; Charness et al., 2013). Recently, preference learning and elicitation of stakeholder preferences has flourished in the context of *human-centric AI* and AI alignment (Jiang et al., 2024; Ji et al., 2023; Capel & Brereton, 2023; Feffer et al., 2023; Kim et al., 2018; Johnston et al., 2023; Sinnott-Armstrong et al., 2021; Liscio et al., 2024; Lee et al., 2019). The usability of learned models in real settings relies on both their predictive accuracy and their alignment to humans' decision-making (Mukherjee et al., 2024; Keswani et al., 2025; Capel & Brereton, 2023; Lima et al., 2021). However, modeling strategies in most prior AI alignment work are agnostic about whether they actually capture human decision-making processes. Our work fills this gap by proposing modeling classes explicitly motivated by the decision rules people report for pairwise comparisons. Like this work, Noothigattu et al. (2020) study the class of binary decision rules arising from pairwise comparisons. However, their axioms concern MLE estimation over data, and thus are *distribution sensitive*, whereas our axioms dictate laws as to how probabilistic preferences must behave *over their domain*. They also do not characterize model classes arising from their axioms, but rather study whether known classes satisfy them, making their work more taxonomically descriptive than prescriptive. Ge et al. (2024) continue this line of inquiry, again with axioms related to estimation from datasets.

There have been other efforts to computationally model human decision-making. Bourgin et al. (2019) propose models bounded by theoretical properties of human decision-making that are fine-tuned with real-world data, but they aim for accurate predictions, rather than cognitive fidelity. Peterson et al. (2021) use neural networks to implement and test cognitive models of participant choices for pairs of gambles. This analysis provides information on the *fit* of various cognitive models, but does not provide a mechanism to *learn* the decision-making process from an individual's data. Plonsky et al. (2017) and Payne et al. (1988) use psychological theories to select feature transformations or simulate prespecified heuristics to obtain gains in predictive power by curating the learning tasks to be more cognitive aligned with human decision processes. However, the applicability of these methods is limited when feature transformations are unknown *a priori* and vary across individuals. Our work also uses relevant works in psychology to identify appropriate decision rule assumptions, but we learn feature transformations from human decisions. Other works use neural networks alongside theory-driven cognitive models to simulate human decision-making (Fintz et al., 2022; Lin et al., 2022), but such models of learned decision processes are largely uninterpretable. Several studies also illustrate that simple heuristic-based models predict human responses well in classification (Holte, 1993; Şimşek & Buckmann, 2015; Brighton, 2006; Czerlinski et al., 1999; Dawes, 1979), supporting our goal of learning heuristics from decisions. Yet, the heterogeneity of heuristics used by different people makes it difficult to find a single model to accurately simulate the responses for an entire population. Our work addresses this issue by learning individual-level decision rules from data.

## 3 A MODEL OF COGNITIVELY-FAITHFUL DECISION-MAKING

In our setting, a decision-maker is presented with a pairwise comparison $(x_1, x_2)$ between two options from the domain $\mathcal{X} \subseteq \mathbb{R}^d$, i.e. each with $d$ descriptive features. For each $i \in [d]$, let $\mathcal{X}_i \subseteq \mathbb{R}$ denote

the input space for feature $i$, such that $\mathcal{X} \doteq \mathcal{X}_1 \times \cdots \times \mathcal{X}_d$. The decision-maker's response function $H : \mathcal{X} \times \mathcal{X} \to [0, 1]$ denotes the probability of choosing the first option $x_1$ for any given pairwise comparison $(x_1, x_2)$. To learn $H$ from observed data, suppose we have a dataset $S$ containing the decision-maker's responses to $N$ pairwise comparisons of the kind $(x_1, x_2, r)$, where $r \in \{0, 1\}$, is the binary decision, with $r = 1$ if the decision-maker deterministically chose $x_1$, and 0 otherwise.

Given dataset $S$, we aim to learn a model $\hat{H}$ that accurately simulates the decision-maker's response function. Taking a general learning approach to this problem, given a hypothesis class of models $\mathcal{H}$, we can search for a model from $\mathcal{H}$ that minimizes the predictive loss over $S$. A common approach to learning $\hat{H}$ is to start with *standard* hypothesis classes, such as classes of linear functions, decision trees, or neural networks. However, as discussed earlier, the assumptions of these classes are likely not faithful to the cognitive processes people truly use to reach their own decisions (Keswani et al., 2025). As such, when using these classes, the resulting model can be difficult to validate and may lead to erroneous predictions in cases where the model class cannot capture the computation required to reach the "correct" decision. We aim to identify a hypothesis class that yields the most accurate model while faithfully representing humans' actual decision-making processes. Such models have the added benefit of being interpretable for human users. To come up with cognitively faithful computational models, we first discuss the computational properties commonly observed in human decision-making processes, and then use these properties to construct an appropriate hypothesis class.

## 3.1 RULE-BASED DECISION-MAKING MODELS

Prior works on cognitive models for human decision-making point to a reliance on *decision rules* and *heuristics* in comparative settings (Gigerenzer et al., 2000; Gigerenzer & Gaissmaier, 2011; Brandstätter et al., 2006; Kahneman & Tversky, 2013; Kahneman & Frederick, 2005). Based on this literature, we construct hypothesis classes that consist of hierarchical decision rules. Our proposed strategy models decision-making for pairwise comparison of options $(x_1, x_2)$ as a two-step process. In the first step, the decision-maker processes each feature in $x_1$ and $x_2$ to *edit* or transform the information presented by this feature. In the second step, the decision-maker combines the edited information from all features to select the *dominant* option. Our hypothesis class will consist of functions that reflect this hierarchical process.

**Editing Rules** The decision rules or heuristic functions used in the first step are referred to as *editing rules*. Editing rules operate at the level of individual features of each element in the given pair and operationalize how a decision-maker processes the given feature value. Let $h^i_{\text{inn}} : \mathcal{X}_i \to \mathcal{X}'_i$ denote the editing rule for feature $i$, with $\mathcal{X}'_i$ the output domain post-editing. Examples of editing include feature simplification (e.g., zeroing out features considered irrelevant), transformation (e.g., log transformation for features with "diminishing returns" or scaling), or leaving the feature unchanged.

While these rules are often *structurally simple*, reflecting their use to reduce cognitive load (Gigerenzer & Gaissmaier, 2011), there can be significant heterogeneity in the editing used in different contexts. For example, prior works note *feature interactions*, where the editing rule used for the $i$th feature $x^{(i)}$ can depend on the values of other features of $x$ (Keswani et al., 2025). To account for this, we will consider the choice of editing rule conditional on the *decision context* features, denoted by the values of features in a set $\omega \subseteq [d]$. On one extreme, $\omega = \emptyset$, implying that each feature is edited independently (i.e., no feature interactions), as $\omega$ grows, model complexity increases as conditional interactions involve more features, and the other extreme, $\omega = [d]$, implies that the choice of editing function for each feature in $x$ can depend on the values of all other features. To account for this context $\omega$, for any option $x \in \mathcal{X}$, we will denote the contextual editing rule operating over feature $i$ by $h^{i, x^{\omega_i}}_{\text{inn}} : \mathcal{X}_i \to \mathcal{X}'_i$, where $\omega_i = \omega \setminus \{i\}$.

**Remark 3.1** (Examples of Editing Rules). *The use of editing rules is well-supported by literature in behavioral economics and social psychology. Montgomery (1983) describes the phase of "separating relevant information from less relevant information which can be discarded," and Kahneman & Tversky (2013) argue that this allows for decision-making based only on the most essential information. In some settings, editing rules represent* feature importance assignment *(Payne, 1976; Gigerenzer et al., 2000). In other cases, editing reflects* feature transformations *to isolate information relevant to the task (Ajzen, 1996; Shah & Oppenheimer, 2008), e.g., thresholding to discretize features, or* log*-transformation to model diminishing returns (Tversky, 1972; Kubanek, 2017).*

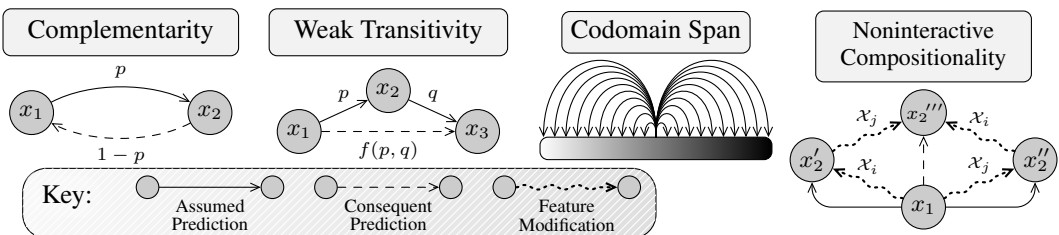

Figure 1: Visualization of the axioms of definition 3.3. Solid arrows represent assumed choice probabilities, dashed arrows represent choice probabilities implied by an axiom, and in *noninteractive compositionality*, dotted arrows represent feature modifications. Arrows may be labeled with choice probabilities.

**Dominance Testing** In the second step of the decision-making process, the edited features are compared across the two options $(x_1, x_2)$ and then combined to reach the final decision on which option is the *dominant one*. We will refer to this second step as the *dominance testing rule*. Let $h_{\text{out}} : \mathcal{X}' \times \mathcal{X}' \to [0, 1]$ denote the dominance test function, where $\mathcal{X}' = \mathcal{X}'_1 \times \cdots \times \mathcal{X}'_d$.

**Remark 3.2** (Examples of Dominance Testing rules)**.** *Several dominance testing rules have been studied in prior literature, including the tallying up heuristic mentioned earlier (Czerlinski et al., 1999; Gigerenzer & Gaissmaier, 2011) and the prominent feature heuristic (choose the element favored by the most prominent feature for which there is non-zero difference in edited feature value) (Persson et al., 2022; Tversky et al., 1988). Alternately, one could create additive or probabilistic functions to capture various dominance testing rules observed in practice, e.g., Bradley-Terry aggregation (1952).*

With this setup, we model a decision maker's response to any pairwise comparison as first applying editing functions $h_{\text{inn}}^{i,x_i}$ over each feature $i$ for both options, given context $\omega$ and $\omega_i = \omega \setminus \{i\}$, and then combining the edited information to reach the final decision using the dominance testing rule $h_{\text{out}}$. Hence, our proposed hypothesis class contains such two-stage models, namely

$$\mathcal{H} \doteq \left\{ x_1, x_2 \mapsto h_{\text{out}} \left( \forall i \in [d], x_1^{(i)} \mapsto h_{\text{inn}}^{i, x_1^{\omega_i}}(x_1^{(i)}), x_2^{(i)} \mapsto h_{\text{inn}}^{i, x_2^{\omega_i}}(x_2^{(i)}) \right) \,\Big|\, h_{\text{inn}}^{\cdot, \cdot} \in \mathcal{H}_{\text{inn}}, h_{\text{out}} \in \mathcal{H}_{\text{out}} \right\} \ .$$

The properties that characterize classes $\mathcal{H}_{\text{inner}}$ and $\mathcal{H}_{\text{outer}}$ are discussed in the next section.

## 3.2 Axiomatic Characterization of Decision Rules

The above-described two-stage process is motivated by extensive decision-making literature. An additional compelling motivation for this hypothesis class comes from an axiomatic characterization of human decision processes for pairwise comparisons. The axioms below describe simple properties expected to be satisfied in an ideal comparative choice model (independent of its functional form).

**Definition 3.3** (Axioms of Binary Choice)**.** *The following axioms describe a binary preference function $H(x_1, x_2) : \mathcal{X}^2 \to \mathcal{Y}$. Unless otherwise stated, each must hold for all $x_1, x_2, x_3 \in \mathcal{X}$.*
*1. **Complementarity**: The order in which two options are presented should not impact the option that is eventually selected. Formally, we require that $H(x_1, x_2) = 1 - H(x_2, x_1)$.*
*2. **Weak Transitivity** (WT): The comparison $H(x_1, x_3)$ can be expressed as a continuous function of the probabilities $H(x_1, x_2)$ and $H(x_2, x_3)$, i.e., for some continuous $f : [0, 1]^2 \to [0, 1]$, it holds that $H(x_1, x_3) = f(H(x_1, x_2), H(x_2, x_3))$.*
*3. **Codomain Span**: For any $p \in (0, 1)$ and $x_1 \in \mathcal{X}$, there exists $x_2 \in \mathcal{X}$ such that $H(x_1, x_2) = p$.*
*4. **Noninteractive Compositionality** (NC): Suppose some $x_1 \in \mathcal{X}$, and $x'_2, x''_2, x'''_2 \in \mathcal{X}$ such that $x'_2$ and $x''_2$ are obtained by changing different features of $x_1$, and $x'''_2$ is produced by applying both changes to $x_1$. We then require that $H(x_1, x'''_2)$ can be computed from $H(x_1, x'_2)$ and $H(x_1, x''_2)$.*
*5. **Conditionally Interactive Compositionality** (CIC): Given a set of condition features $\omega \subseteq [d]$, suppose some $x_1 \in \mathcal{X}$, and $x'_2, x''_2, x'''_2 \in \mathcal{X}$ such that $x'_2$ and $x''_2$ are obtained by changing different features (neither in $\omega$) of $x_1$, and $x'''_2$ is produced by applying both changes to $x_1$. We then require that $H(x_1, x'''_2)$ can be computed from $H(x_1, x'_2)$ and $H(x_1, x''_2)$.*

These axioms are visualized in fig. 1. *Complementarity* (1) ensures that the order in which options are presented should not matter, and predicted probabilities sum to 1. *Weak transitivity* (2) requires that comparisons between $(x_1, x_2)$ and $(x_2, x_3)$ carry all the information about the items $x_1, x_2, x_3$ to also compare $(x_1, x_3)$, i.e., the first two pairwise comparisons suffice to "complete the triangle" of all

three pairwise comparisons. *Codomain span* (3) is a technical condition, essentially describing a type of continuity, as to specify the decision rule's transitivity law for *all possible probabilities*, we must ensure that all possible probabilities *actually arise* in comparisons over the domain $\mathcal{X}$. Theorem 3.4 item 2 shows that these properties work together to induce a strong form of transitivity in probabilistic predictions, and the discrete form given as item 1 relaxes the *codomain span* (3) assumption.

Axioms 4 & 5 characterize the extent to which different features interact in the decision process. *Noninteractive compositionality* encodes a form of *feature independence*, requiring that the impact of changing two different features is compositional, which is less restrictive than assuming linearity, but restricts interactions between features in a similar manner. Essentially, this axiom characterizes a *generalized additive model* (GAM) (Hastie, 2017), as it dictates how information is synthesizes *across features*. *Conditional interactivity* then generalizes this idea, allowing for non-additive *conditional feature interactions*. We propose this axiom to account for a larger class of models that consider the decision context, where the impact of changing two features is additively compositional, *except if* one of the features is a *condition feature* (part of the *context $\omega$* described in section 3.1).

Axioms 1–3 are prescriptive, describing properties that may be considered desirable by many, whereas 4–5 are not assumed to hold universally (potentially varying even within an individual), but rather they suffice characterize a class of simple decision-making rules when they do hold. This stands in contrast to the strong assumptions made by classical axiomatic models of decision making. Moreover, they are agnostic to the functional form of the decision rule and do not assume linear utility, stable preferences, or optimality.

We emphasize that our axioms are strictly weaker than those commonly used in classical choice theory. For example, *WT* does not assume that preferences are globally transitive or derivable from a single latent utility function; it merely requires that a comparison between two options can be constructed from intermediate comparisons. Similarly, *NC* is weaker than additive utility or independence of irrelevant alternatives: it constrains how feature modifications combine without assuming linearity or ruling out contextual dependence. *CIC* further relaxes this requirement by explicitly allowing structured feature interactions. As a result, our framework accommodates heuristic and context-sensitive decision processes documented in empirical studies, while still enabling formal analysis and learnability guarantees. We now show that the decision-making processes that satisfy these axioms follow the two-stage process outlined in section 3.1.

**Theorem 3.4** (Axiomatic Factoring Characterization). *We now explore the consequences of our axioms on characterizing the response function $H(\cdot, \cdot)$ of a decision-maker.*
*1. **Atomic Model** Suppose axiom 1, and also that $\mathcal{X}$ is a countable domain. Then there exists an* atomic rule $h_{inn} : \mathcal{X} \to \mathbb{R}$ and $h_{out} : \mathbb{R} \to [0, 1]$ *s.t. $H(\cdot, \cdot)$ can be factored as*

$$H(x_1, x_2) = h_{out}\left(h_{inn}(x_1) - h_{inn}(x_2)\right) \ .$$

*2. $\sigma(\cdot)$-**Transitivity** Suppose axioms 1–3 (Complementarity, WT, and Codomain Span) hold. Then there exists a symmetric continuous random variable with full support and CDF $\sigma(\cdot)$ s.t.*

$$H(x_1, x_3) = \sigma\left(\sigma^{-1}(H(x_1, x_2)) + \sigma^{-1}(H(x_2, x_3))\right) \ .$$

*Moreover, there exists some* atomic rule $h_{inn} : \mathcal{X} \to \mathbb{R}$ *such that $H(\cdot, \cdot)$ can be factored as*

$$H(x_1, x_2) = \sigma\left(h_{inn}(x_1) - h_{inn}(x_2)\right) \ .$$

*3. **Unconditional Factor Model**: Suppose axioms 1–3 and 4 (NC). Then there exist inner functions $h_{inn}^i : \mathcal{X}_i \to \mathbb{R}$ for $i \in [d]$ such that the atomic rule $h_{inn}(\cdot)$ and decision rule $H(\cdot, \cdot)$ factor as*

$$h_{inn}(x) = \sum_{i=1}^{d} h_{inn}^i(x^{(i)}) \ , \quad H(x_1, x_2) = h_{out}\left(\sum_{i=1}^{d} h_{inn}^i(x_1^{(i)}) - h_{inn}^i(x_2^{(i)})\right) \ .$$

*4. **Conditional Factor Model**: Suppose axioms 1–3 and 5 (CIC). Then there exist conditional inner functions $h_{inn}^{i, \cdot} : \mathcal{X}_i \to \mathbb{R}$ for $i \in [d]$ and condition features (context) $\omega \subseteq [d]$, $\omega_i = \omega \setminus \{i\}$, s.t.*

$$H(x_1, x_2) = h_{out}\left(\sum_{i=1}^{d} h_{inn}^{i, x_1^{\omega_i}}(x_1^{(i)}) - h_{inn}^{i, x_2^{\omega_i}}(x_2^{(i)})\right) \ .$$

Item 1 shows that axiom 1 suffices to reduce binary choices to a function of a difference of atomic predictions for continuous transitivity laws $f(\cdot, \cdot)$, i.e., to $H(x_1, x_2) = h_{\text{out}}(h_{\text{inn}}(x_1) - h_{\text{inn}}(x_2))$, thus recovering the two-stage modeling process described in section 3.1. However, there is little

structure beyond this: Such a factoring *exists*, but it is by no means *unique*, and it may well be *intransitive*. Item 2 then imposes weak transitivity through axiom 2, additional *continuity structure* through axiom 3, and paired with continuity of the transitivity law $f(\cdot, \cdot)$, we may conclude that $h_{\text{out}}(\cdot)$ is a sigmoid function (CDF). Axioms 4–5 are then needed to control feature interactions, and dictate a two-stage model structure wherein each feature is processed individually or conditionally.

Intuitively, with $\sigma$-transitivity, we require that transitive probabilities can be computed via addition within the sigmoid. Therefore, transitivity along chains of probabilities above $\frac{1}{2}$ grow ever closer to 1 as $\sigma(\cdot)$ is applied to a growing sum. In the parlance of generalized linear models, $\sigma^{-1}$ plays the role of a link function. Bradley-Terry models, such as logistic regressors, employ the logistic sigmoid, but other sigmoids, such as the Gaussian CDF (with probit link function) also see use, e.g., in GLMs (McCullagh & Nelder, 1989).

We now describe *strong assumptions* that may only be appropriate in specific real-world domains, but nevertheless produce specific properties for the hypothesis class of editing function $\mathcal{H}_{\text{inner}}$. Unlike our axioms of definition 3.3, these assumptions are highly situational, and they describe behavior that may be considered reasonable in some circumstances, but their absence should rarely be viewed as surprising or undesirable in and of itself.

**Definition 3.5** (Assumptions on Binary Preference Models)**.** *We state assumptions on $H$ that are suitable for discrete or continuous domains. These discrete properties must apply to all $x_1, x_2 \in \mathcal{X}$.*
*1. $\sigma$-Linearity: There exists some $\boldsymbol{w} \in \mathbb{R}^d$ such that $\sigma^{-1} \circ H(x_1, x_2) = \boldsymbol{w} \cdot (x_1 - x_2)$.*
*2. Monotonicity: If $x_1 \preceq x_2$, then $H(x_1, x_2) \leq \frac{1}{2}$.*

We now combine these stronger assumptions with the results of the axiomatic analysis of theorem 3.4. Recall that our basic axioms imply the factoring

$$\mathcal{H} = \{x_1, x_2 \mapsto \sigma(h_{\text{inn}}(x_1) - h_{\text{inn}}(x_2)) \,|\, h_{\text{inn}} \in \mathcal{H}_{\text{inn}}\} \ .$$

**Theorem 3.6** (Axiomatic Models)**.** *Suppose axioms 1–4. The following restrict $\mathcal{H}_{inn}$ or each $\mathcal{H}_{inn}^{(i)}$.*
*1. Suppose $\sigma$-linearity. Then $\mathcal{H}_{inn}^{(i)} = \{x \mapsto wx \,|\, w \in \mathbb{R}\}$, thus $\mathcal{H} = \{x_1, x_2 \mapsto \sigma(\boldsymbol{w} \cdot (x_1 - x_2)) \,|\, \boldsymbol{w} \in \mathbb{R}^d\}$.*
*2. Suppose monotonicity. Then $\mathcal{H}_{inn}^{(i)} = \{h : \mathbb{R} \to \mathbb{R} \,|\, x \leq y \implies h(x) \leq h(y)\}$.*
*3. Multivariate Monotonic Models: If we assume monotonicity but relax noninteractive compositionality, then $\mathcal{H} = \{x_1, x_2 \mapsto \sigma(h(x_1) - h(x_2)) \,|\, h : \mathcal{X} \to \mathbb{R} \text{ s.t. } \vec{x} \preceq \vec{y} \implies h(\vec{x}) \leq h(\vec{y})\}$.*
*4. Conditional GAM Tree If we relax NC to CIC, then $h_{inn}(\cdot)$ may be represented as a hybrid model of a decision tree / GAM model, starting with a tree over $\mathcal{X}_\omega$, where each leaf contains a GAM over $\mathcal{X}_{\backslash\omega}$. Formally, we have $\mathcal{H}_{inn} = \left\{ \sum_{i=1}^{d-|\omega|} \left(T_i(x^\omega)\right) \left((x^{\backslash\omega})^{(i)}\right) \,\big|\, T \in \mathbb{R}^{|\omega|} \to (\mathbb{R} \to \mathbb{R})^{d-|\omega|} \right\}$.*

From this perspective, we derive a few valuable insights: *Logistic regression* with nonnegative weights is abstractly characterized by Bradley-Terry aggregation ($\sigma(u) = \text{logistic}(u)$) and linearity (and implicitly implies noninteractivity and monotonicity). If linearity is relaxed to monotonicity, we obtain the class of univariate monotonic models, and subsequently when noninteractivity is relaxed, we obtain univariate monotonic models with conditional interactions. Similarly, *probit regression* is characterized by taking $\sigma(u) = \Phi(u)$, i.e., the Gaussian CDF function, and linearity.

**Extending Pairwise Comparisons to $n$-Way Choice** We focus on binary preference elicitation because it is a well-studied task in the decision-making literature. To extend our binary preference framework to learn probabilistic rankings over $n$ items, we use the *choice axiom* of Luce et al. (1959): *Introducing additional items should not change the probability ratio of choosing $x_1$ to choosing $x_2$,* Without WT, there are $\binom{n}{2}$ degrees of freedom (DoF) to complementary binary preferences, and a probabilistic ranking that accords with these binary preferences in general does not exist (e.g., preferences over rocks-paper-scissors cannot accord with the choice axiom). However, with WT, there are only $n - 1$ DoF, as a tree of predictions spanning $n$ items can be completed via transitivity.

**Theorem 3.7** (Proportionality of Choice)**.** *Suppose a probabilistic decision rule over $n \geq 3$ items that obeys the choice axiom. Then $H(\cdot, \cdot)$ admits a $\sigma$-transitive factoring with logistic $\sigma(\cdot)$.*

Theorem 3.7 uniquely characterizes the Bradley-Terry model, but is only interesting insofar as the model is believable. Alternative models of $n$-way decision making are also well-studied, such as noisy observation, where utility + noise is observed for each option, and the largest observation wins, resulting in a probability distribution over outcomes. In particular, the Bradley-Terry model arises (non-uniquely) from homoskedastic Gumbel noise, but of course the Thurstone–Mosteller model

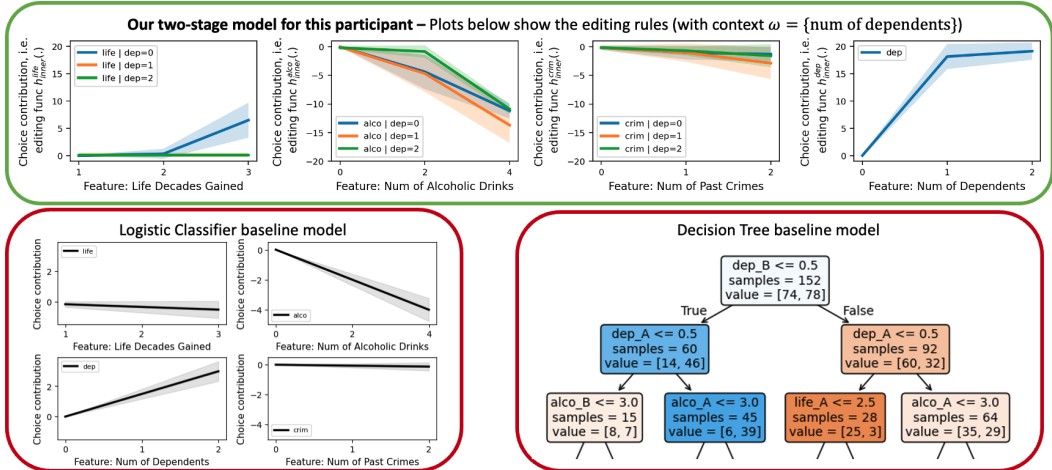

Figure 2: Interpretable models learned for P4 of Study One. For our two-stage model and logistic classifier, we plot the contributions of each value of each feature to the final choice i.e., we plot $h_{\text{inn}}(\cdot)$. We also show the decision tree, truncated to the first three layers. Note the differences in model interpretation between these strategies, with our model providing insight into the heuristics used by P4.

(Thurstone, 1927; Mosteller, 1951) arises from homoskedastic Gaussian noise (hence the probit link function), and from this perspective, $\sigma$-transitivity can be generalized to $n$-way decision models whenever the distribution with CDF $\sigma$ decomposes as the difference of two i.i.d. random variables, i.e., is in the *difference-form decomposable* (DFD) family, although such decompositions *are not always unique* (Carnal & Dozzi, 1989; Ewerhart & Serena, 2025).

Overall, with reasonable domain-specific assumptions, we recover other standard modeling classes from our two-stage modeling framework, highlighting the expressive power of our modeling class. Importantly, assumptions like linearity and monotonicity are domain-dependent and need to be qualitatively justified for the problem context. Classes, like logistic regressors, that implicitly encode these assumptions may not generate cognitively-faithful models when the human decision process violates these assumptions. Instead, we based our modeling class on generic axioms of human decision processes, which can be further constrained by domain-specific assumptions as necessary.

# 4 EMPIRICAL ANALYSIS ON KIDNEY ALLOCATION DATA

We test our modeling strategy to learn computational models of people's moral judgments regarding the allocation of kidneys to patients based on their medical attributes (e.g., transplant outcomes) and non-medical attributes (e.g., dependents and lifestyles) (Boerstler et al., 2024; Chan et al., 2024). Prior works have noted several limitations of existing modeling approaches in this domain (Keswani et al., 2025), making this is a relevant case study to analyze our proposed framework.

**Real-world dataset** We use the dataset of Boerstler et al. (2024), which spans two studies where participants were presented with several kidney allocation scenarios (15 participants in Study One and 40 in Study Two). In both studies, each kidney allocation scenario comprises a pairwise comparison between two patients (see fig. 4 for an example). Participants were instructed to choose which patient should be given an available kidney. In Study One, the patient features presented are the patient's *number of dependents*, projected *life years gained* from the transplant (LYG), *alcoholic drinks per day*, and *number of crimes committed*. In Study Two, the patient features are the patient's *number of elderly dependents*, LYG, *years waiting* for the transplant, *weekly work hours post-transplant*, and *obesity level*. We present further details in appendix B.

**Synthetic dataset** We also created a synthetic dataset representing multiple simulated decision-makers. This dataset contains pairwise comparisons of hypothetical kidney patients described using four features: *number of dependents*, LYG, *years waiting*, and *number of crimes* committed. The goal is to simulate decision-makers who explicitly use the heuristics observed in prior studies to assess how well our model and other baselines recover these heuristics. The heuristics are taken from Keswani et al. (2025), who provide user-reported qualitative accounts of decision rules people use

for kidney allocation decisions. Using their observations, we create five simulated decision-makers, DM1–DM5, each using different decision rules over the presented patient features. For instance, DM1 decides between A and B with the following process: (a) assign 1 point to the patient with a higher LYG; (b) assign 1 point to each patient with dependents; (c) assign 1 point to each patient on the waiting list for $>6$ years; (d) add $\mathcal{N}(0, 1)$ noise (i.e., a homoskedastic Thurstone-Mosteller process) to the difference of patient scores, and choose A if difference$> 0$ and B otherwise (DM1's process is mathematically described in fig. 3). Other simulated decision-makers (DM2–DM5) also use various heuristics, such as *thresholding*, *diminishing returns*, and *tallying* (described in appendix B). Our dataset contains 1000 pairwise comparisons from each simulated DM.

**Methodology and Baselines** We compare our model to several learning approaches, namely Bradley-Terry and Drift Diffusion models from cognitive science literature, and common supervised learning approaches, such as logistic and elastic-net classifiers, SVM, GAM with spline terms, decision trees, multi-layer perception (MLP), $k$-NN, and random forests. Drift Diffusion is the only model we consider that requires reaction times per scenario, thus it is not run for the simulated DMs. We use logistic models and decision tree models as the "interpretable" methods to compare our model's interpretability to. Since moral judgments vary substantially across individuals, all experiments operate over individual-participant-level data. For each decision-maker (real or simulated), we use a 70-30 train-test split, and report test accuracy over 20 repetitions. For our framework, we minimize the predictive loss to learn the editing functions $h_{\text{inn}}^{\cdot\cdot}$ (which assign score$\in \mathbb{R}$ to each feature value), constraining all $h_{\text{inn}}^{\cdot\cdot}$ to be monotonic (since all features in our dataset can be seen to impact the final choice monotonically) We implement two variants of this framework: (A) with cross-entropy loss and $\sigma(x) = (1 + e^{-x})^{-1}$, and (B) with hinge loss and $\sigma$ as the identity function. The first variant is aligned with the probabilistic framework of theorem 3.4, while the second framework is better suited to assessing our learned model on the "hard classification" metric of 0-1 predictive accuracy. Context $\omega$ is limited to one feature for the real-world datasets, chosen using cross-validation, and $\emptyset$ for the synthetic dataset. Appendix B provides additional implementation details.

**Results** Across all datasets, we observe that our cognitively-motivated models achieve high accuracy and provide deeper insight into decision-making processes than other baselines.

*Aggregated Performance.* The mean accuracy of each model over all participants is shown in table 1. Overall, on both real and synthetic data, our framework produces models *at least as accurate* as all baselines. Individual-level performance is presented in appendix B. Additionally, editing functions $h_{\text{inn}}$ learned for the decision-makers provide insight into how they process the input features to reach their final decision. We demonstrate this interpretability through case studies of the editing functions learned for the simulated decision-maker DM1 and a real-world participant P4 from Study One.

*Participant P4 in Study One.* The two-stage model learned for P4 (w/ hinge loss) is illustrated in fig. 2 (top row). The editing function plots show the

| Model | Average Accuracy (stddev) | | |
| --- | --- | --- | --- |
| | Study One | Study Two | Simulated |
| **Cognitive models** | | | |
| Drift-Diffusion | .89 (.05) | .88 (.05) | – |
| Bradley-Terry | **.90** (.06) | .78 (.06) | .77 (.06) |
| **Supervised learning models** | | | |
| Logistic Clf | **.90** (.06) | .89 (.05) | .85 (.07) |
| Elastic Net | .89 (.04) | .88 (.05) | .85 (.07) |
| SVM | .89 (.06) | .89 (.05) | .85 (.07) |
| GAM | .87 (.09) | .84 (.11) | .88 (.08) |
| Decision Tree | .83 (.06) | .79 (.06) | .82 (.11) |
| k-NN | .85 (.06) | .82 (.05) | .79 (.08) |
| MLP | .89 (.05) | .86 (.06) | .87 (.08) |
| Random Forest | .86 (.05) | .85 (.04) | .87 (.08) |
| **Our Models** | | | |
| cross-entropy loss | **.90** (.06) | **.90** (.05) | **.89** (.10) |
| hinge loss | **.90** (.06) | .89 (.06) | **.89** (.08) |

Table 1: Performance statistics of all individualized models on kidney allocation datasets.

following nuances of their decision-making process for pairwise comparisons of kidney patients: (a) *number of dependents* and *number of alcoholic drinks* are the two most important features, as reflected by the large choice contributions of these features; (b) *number of past crimes* is considered mostly irrelevant by this participant; (c) *life decades gained* is relevant only when the patient has zero dependents, indicating a conditional interaction between these two features; (d) for *number of dependents*, a difference of 1 vs 0 dependents is much more significant to the final choice than a difference of 2 vs 1 dependents — approximately a threshold decision rule such that any non-zero number of dependents contributes equally; (e) similarly for *life decades gained* and *number of alcoholic drinks*, for certain conditions on *number of dependents*, the participant employs threshold decision rules. Our model is able to learn all of these decision-making nuances with an accuracy of 0.78 ($\pm$ 0.05), but a trained logistic classifier (bottom-left in fig. 2) has a lower accuracy of

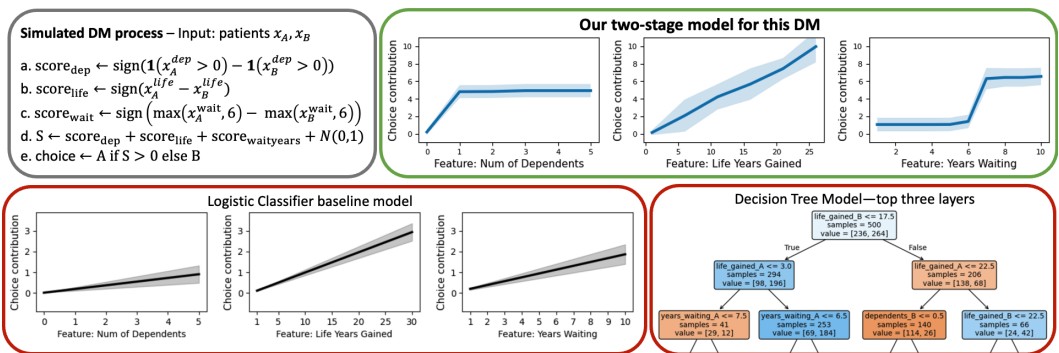

Figure 3: Models learned for the simulated decision-maker DM1. On the top left, we present the mathematical description of the DM1's process. Once again, we present each model's interpretation of the process used by the decision-maker. Observe that our two-stage model most accurately captures the simulated process.

0.76 ($\pm$ 0.04) and only uncovered points (a) and (b) above, while a trained decision-tree model (bottom-right in fig. 2) has an accuracy of 0.70 ($\pm$ 0.03) and only uncovered points (a), (c), and (d).

*Simulated DM1.* Figure 3 (top right) shows how our model recovers the rules used by DM1. E.g., for the feature *number for dependents*, $h_{\text{inn}}^{\#\text{deps}}$ captures the increase in choice contribution when this feature has a non-zero value, and for the *years waiting* feature, $h_{\text{inn}}^{\text{wait yrs}}$ learns the increased contribution of values exceeding 6 years. Both successfully represent the threshold functions used in DM1. Note that neither of these observations can be easily inferred from logistic or decision tree models (bottom row of fig. 3). Our model is also more accurate ($0.76 \pm 0.02$) than logistic ($0.74 \pm 0.02$) and decision tree ($0.68 \pm 0.04$) models. Appendix B shows similar results for other simulated decision-makers. Overall, our models provide a better understanding of the decision-making processes, without sacrificing predictive accuracy.

## 5    DISCUSSION, LIMITATIONS, AND FUTURE WORK

This work provides a modeling strategy to learn human decision processes from pairwise comparisons. The primary goal of this strategy is *fidelity*, such that best-fit models from these classes are more faithful to humans' actual decision-making than standard hypothesis spaces. Our theoretical analysis demonstrates that these classes emerge from natural decision-making axioms, and our empirical analysis shows that greater *fidelity* would lead to greater model *accuracy* in some cases. The advantages of cognitively-faithful models will depend on the application. In applications like online recommender systems, it may be sufficient to predict human behavior accurately, without simulating their actual decision processes. However, cognitively-faithful models would be desirable in other high-stakes AI domains (e.g., healthcare and sentencing), where the interpretability and alignment of AI's decisions with the users is pivotal in establishing trust (Jacovi et al., 2021). Cognitively-faithful models would also be preferable in domains with no "ground truth" (like kidney allocation), where model validation requires the user to understand and evaluate the model's decision process. Lastly, cognitively-faithful models are essential for any use of AI in moral domains, since stakeholders expect an AI to justify its moral decisions in a similar manner as humans (Lima et al., 2021).

Our mathematical model is motivated by human behavior, but due to the complexity of the latter, any such effort is inherently limited. The interpretability features of our framework need further validation through real-world user studies. While this work provides a proof-of-concept for a computational framework to learn from human decisions, future user studies can help assess the extent to which users understand and trust the learned two-stage models. Our axioms characterize the "ideal" decision-making process of an individual; i.e., the process they would prefer to follow in the absence of any internal/external constraints, or the process that describes their judgments about what "should be done." However, human decisions are known to deviate from the ideal in many ways, e.g., sometimes exhibiting transitivity and complementarity violations (Tversky et al., 1990), among others. One thus wonders, does precluding such deviations aligns an AI system to the processes humans actually use, or instead to some idealized version thereof? Both have their virtues, but we must be aware of the differences. We hope that future work can study this issue, and explore ways to extend our two-stage hypothesis classes to quantify deviations of one's "ideal" judgments from their implemented choices.

## ACKNOWLEDGMENTS

VK, CC, JSB, and WSA are grateful for the financial support from OpenAI and Duke University.

HH acknowledges support from the CMU-NIST Cooperative Research Center on AI Measurement Science & Engineering (AIMSEC), and the AI Research Institutes Program funded by the National Science Foundation under AI Institute for Societal Decision Making (AI-SDM), Award No. 2229881. Any opinions, findings, conclusions, or recommendations expressed in this material are those of the authors and do not reflect the views of the funding agencies.

## REPRODUCIBILITY STATEMENT

For theoretical results, full proofs of all results are provided in Appendix A. To facilitate reproducibility of the empirical results, the primary methodological details of our simulations are described in Section 4, and additional descriptions, including simulated DM construction processes, training procedures, and hyperparameter values, are given in Appendix B. Additionally, source code for these simulations is available at `https://github.com/vijaykeswani/Cognitively-Faithful-Decision-Models/`.

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
