*Proof.* The structure of this result is a bit convoluted, due to its presentation in four parts. We first show item 1 in isolation using a straightforward explicit construction. We then show that the conditions of item 2 imply *symmetry* of the transitivity law, and the assumed addition of *continuity* of $f(\cdot, \cdot)$ is sufficient to yield item 2.

From there, we address items 3 & 4. Note that because NC is a special case of CIC with condition set $\omega = \emptyset$, we show only item 4, as item 3 is a special case of item 4.

We first show item 1. For countable $\mathcal{X}$, it's trivially true that each $x \in \mathcal{X}$ can be encoded as a power of 2. Then, $h_{inn}(x_1) - h_{inn}(x_2)$ is then 0 iff $x_1 = x_2$, which by axiom 1, necessitates $h_{out}(0) = \frac{1}{2}$. Otherwise, $x_1 \neq x_2$, and we can recover $x_1$ and $x_2$ uniquely (the larger by rounding the absolute difference up to the nearest power of 2, i.e., by taking $\log_2(|h_{inn}(x_i) - h_{inn}(x_j)|)$, the smaller via subtraction, and the sign indicating which is which). Therefore, using this construction of $h_{inn}$, for an arbitrary mapping between $\mathcal{X}$ and $\mathbb{Z}$, we can construct a corresponding $h_{out}$ that reconstructs any possible decision rule $H(\cdot, \cdot)$ over $\mathcal{X}$.

We now show item 2.

We first show that these assumptions imply $f(u, v) = f(v, u)$ for all $u, v \in (0, 1)$, i.e., the transitivity law is symmetric. Assume WLOG $u \geq \frac{1}{2}$ and $v \geq \frac{1}{2}$, as by complementarity and WT, the remaining cases can be derived. Now, select an arbitrary $x_0$ in $\mathcal{X}$. For any $\varepsilon \in (0, \frac{1}{2})$, by *codomain span*, there exists an infinite sequence $x_1, x_2, \ldots$ such that $H(x_{i-1}, x_i) = \frac{1}{2} + \varepsilon$, and moreover, let $k_1, k_2$ be the smallest integers such that $H(x_0, x_{k_1}) \geq u$ and $H(x_0, x_{k_1}) \geq v$, respectively. Note that for any $k$, $H(x_0, x_k)$ is an increasing function, achieving $\lim_{k \to \infty} H(x_0, x_k) = 1$. Furthermore, by continuity, in the limit as $\varepsilon \to 0$, $H(x_0, x_{k_1})$ and $H(x_0, x_{k_2})$ converge to $u$ and $v$, respectively. However, the grouping of this sequence into a segment of $k_1$ points approximating $u$ and $k_2$ points approximating $v$ was arbitrary, and the weak transitivity operator $f(\cdot, \cdot)$, by construction, must be associative. We may thus conclude that, due to continuity and the limit as $\varepsilon \to 0$, $f(u, v) = f(v, u)$.

Now observe that, by the continuous deep sets theorem (Theorem 7 of Zaheer et al. (2017)), since $f(\cdot, \cdot)$ is symmetric, there exist continuous functions $\rho, \phi : \mathbb{R} \to \mathbb{R}$ such that

$$H(x_1, x_3) = f(H(x_1, x_2), H(x_2, x_3)) = \rho(\phi \circ H(x_1, x_2) + \phi \circ H(x_2, x_3)) \ .$$

Assume WLOG that $\phi(\frac{1}{2}) = 0$ (as any shifting to $\phi$ can be absorbed by $\rho$ in the composition).

Complementarity implies reflexivity, i.e., it holds that $H(x_1, x_1) = 1 - H(x_1, x_1) = \frac{1}{2}$. Consequently, we observe the f Now, again by reflexivity, and applying our derived weak transitive law, it holds

$$H(x_1, x_2) = \rho(\phi \circ H(x_1, x_2) + \phi \circ H(x_2, x_2))$$

$$= \rho(\phi \circ H(x_1, x_2)) \qquad\qquad \phi \circ H(x_2, x_2) = \phi(\frac{1}{2}) = 0 \ .$$

Thus $\rho(\phi(u)) = u$ for all $u \in (0, 1)$.

Now, we argue that $\phi(u) + \phi(1 - u) = 0$, which implies $\phi(u) = -\phi(1 - u)$. Observe the following cancellation: $H(x_1, x_1) = f(H(x_1, x_2), H(x_2, x_1)) = \rho(\phi(u) + \phi(1 - u)) = \rho(0) = \frac{1}{2}$. By continuity, and the property $\rho(\phi(u)) = u$, this implies that $\phi$ is either an increasing or a decreasing function. Assume now that, WLOG, $\phi$ (and consequently $\rho$) are both increasing functions, as output-negation of $\phi$ may be counterbalanced by input-negation of $\rho$.

To show that $\rho$ and $\phi$ are true inverses, we require also that $\phi(\rho(v)) = v$ for all $v \in \mathbb{R}$. We show this by proving that $\rho(\cdot)$ is a strictly increasing continuous function on $\mathbb{R} \to (0, 1)$. Because $\phi$ is a strictly increasing function, and $\rho(\phi(1)) = 1$, clearly $\rho(\infty) = 1$ and $\rho$ is a strictly increasing continuous function *on the domain* $(\phi(0), \phi(1))$. The only issue is that, if $\phi(1) = -\phi(0) < \infty$, then $\rho$ is only *weakly increasing* (constant) over the rest of $\mathbb{R}$. Now, suppose BWOC that $\phi(1) = c$ for some $c < \infty$. By codomain span and increasingness of $\rho$, there exist $x_1, x_2, x_3, x_\infty$, such that $H(x_1, x_2) \neq H(x_1, x_3)$, $H(x_i, x_\infty) = 1$ for each $i$, and the weak transitivity law is violated, as $H(x_1, x_2)$ and $H(x_1, x_3)$ *can not possibly* both be computed as $f(1, 1)$. NB this impossibility does not depend on the functional form we derive: Rather, there is a fundamental incompatibility between weak transitivity and prediction with certainty. We thus conclude that the image of $\phi(\cdot)$ is $\mathbb{R}$, hence $\phi$ and $\rho$ are true inverses, and both strictly monotonically increasing continuous functions. Moreover, $\rho$ takes the form of the CDF of a symmetric continuous random variable with support $\mathbb{R}$.

Henceforth, we have operated purely in terms of the predicted probabilities, i.e., $H(\cdot, \cdot)$, but the result requires us to draw conclusions in terms of some $h_{\text{inn}}(x)$. We now argue for the existence of such a decomposition. Essentially, we "eliminate the middle man," as

$$\begin{aligned}
H(x_1, x_3) &= f(H(x_1, x_2), H(x_2, x_3)) && \text{WEAK TRANSITIVITY} \\
&= \rho(\phi \circ H(x_1, x_2) + \phi \circ H(x_2, x_3)) && \text{SEE ABOVE} \\
&= \rho(\phi \circ H(x_1, x_2) - \phi \circ H(x_3, x_2)) && \text{COMPLEMENTARITY} \\
&= \rho\left((h_{\text{inn}}(x_1) - h'_{\text{inn}}(x_2)) - (h_{\text{inn}}(x_3) - h'_{\text{inn}}(x_2))\right) && \text{SEE BELOW} \\
&= \rho\left(h_{\text{inn}}(x_1) - h_{\text{inn}}(x_3)\right) \ .
\end{aligned}$$

The step marked SEE BELOW is rather subtle, but observe that it must hold BWOC for some functions $h_{\text{inn}}, h'_{\text{inn}} : \mathcal{X} \to \mathbb{R}$, as if it did not, then there would exist some $x_2$, $x'_2$ such that $H(x_1, x_3) = f(H(x_1, x_2), H(x_2, x_3)) \neq f(H(x_1, x'_2), H(x'_2, x_3))$, which violates weak transitivity. Essentially, due to invertibility, $\phi \circ H(x_1, x_3) = (h_{\text{inn}}(x_1) - h'_{\text{inn}}(x_2)) - (h_{\text{inn}}(x_3) - h'_{\text{inn}}(x_2))$ must be invariant under choice of $x_2$. In the subsequent step, $h'_{\text{inn}}$ is eliminated, which effectively illustrates that $h_{\text{inn}} = h'_{\text{inn}}$. Finally, let $h_{\text{out}}(u) = \phi(u)$, and item 1 is complete.

We now show item 4 (noting again that item 3 is a special case).

We first show a technical lemma: we assume $x'''$ can be obtained from $x_1$ by changing two independent features not in the condition set $\omega$, each partial change resulting in $x'_2$ and $x''_2$ (as in the CIC axiom). Observe that

$$\sigma^{-1} \circ H(x_1, x'''_2)$$

$$= \sigma^{-1} \circ H(x_1, x'_2) + \sigma^{-1} \circ H(x'_2, x'''_2) + \sigma^{-1} \circ H(x_1, x''_2) + \sigma^{-1} \circ H(x''_2, x'''_2) - \sigma^{-1} \circ H(x_1, x'''_2) \quad \text{$\sigma$-TRANSITIVITY}$$

$$= \left(\sigma^{-1} \circ H(x_1, x'_2) + \sigma^{-1} \circ H(x_1, x''_2)\right) + \left(\sigma^{-1} \circ H(x'_2, x'''_2) + \sigma^{-1} \circ H(x''_2, x'''_2) + \sigma^{-1} \circ H(x''_2, x_1)\right) \quad \text{ALGEBRA}$$

$$= \left(\sigma^{-1} \circ H(x_1, x'_2) + \sigma^{-1} \circ H(x_1, x''_2)\right) +$$
$$\frac{1}{2}\left(\sigma^{-1} \circ H(x'_2, x'''_2) + \sigma^{-1} \circ H(x''_2, x'''_2) + \sigma^{-1} \circ H(x'_2, x_1) + \sigma^{-1} \circ H(x''_2, x_1)\right) \quad \text{ALGEBRA}$$

$$= \frac{1}{2}\left(\sigma^{-1} \circ H(x_1, x'_2) + \sigma^{-1} \circ H(x_1, x''_2)\right) + \frac{1}{2}\left(\sigma^{-1} \circ H(x'_2, x'''_2) + \sigma^{-1} \circ H(x''_2, x'''_2)\right) \quad \text{ALGEBRA}$$

$$= \sigma^{-1} \circ H(x_1, x'_2) + \sigma^{-1} \circ H(x_1, x''_2) \quad \text{SEE BELOW}$$

The final step applies the assumed CIC axiom: The only way this assumption can hold is if $\left(\sigma^{-1} \circ H(x_1, x_2') + \sigma^{-1} \circ H(x_1, x_2'')\right) = \left(\sigma^{-1} \circ H(x_2', x_2''') + \sigma^{-1} \circ H(x_2'', x_2''')\right)$. This is because of symmetry, if we negate the equation, flipping $x_1$ and $x_2'''$, the same must hold, hence the conclusion of the technical lemma.

We now chain the above result to derive the desideratum, i.e., item 2. Given $x_0', x_1', x_2', x_3', \ldots, x_d'$, define the *transitive chain operator* $f(\cdot, \cdot, \cdots, \cdot)$ as

$$f(x_0', x_1', x_2', x_3', \ldots, x_d') = f(\cdots (f(f(x_0', x_1'), x_2'), \cdots), x_d') = h_{\text{out}}\left(\sum_{i=1}^{d} h_{\text{inn}}(x_{i-1}') - h_{\text{inn}}(x_i')\right) \ ,$$

where the RHS applies the structure of $\sigma$-transitivity.

Now, take $x_\omega'^{i}$ to be $x_2^i$ if $i \in \omega$, $x_1^i$ otherwise. Define a sequence $x_j'$ such that each $x_j'$ takes the previous $x_{j-1}'$ (starting with $x_\omega'$ for $j = 1$) and changes one feature $k \notin \omega$ from $x_1^k$ to $x_2^k$, such that $x_{d-|\omega|}' = x_2$.

We then chain the result over all items not in the condition set $\omega$ to obtain

$$
\begin{aligned}
H(x_1, x_2) &= h_{\text{out}}\left(h_{\text{inn}}(x_1) - h_{\text{inn}}(x_2)\right) && \text{BY ASSUMPTION} \\
&= f(x_1, x_{2,\omega}, x_{2,(1)}, x_{2,(2)}, \ldots, x_{2,(d-|\omega|)}) && \text{TRANSITIVITY CHAIN} \\
&= h_{\text{out}}\left(h_{\text{inn}}^{\omega, x_1^\omega}(x_1) - h_{\text{inn}}^{\omega, x_2^\omega}(x_2) + \sum_{i \notin \omega} h_{\text{inn}}^{i, \omega_i}(x_1^i) - h_{\text{inn}}^{i, \omega_i}(x_2^i)\right) && \sigma\text{-TRANSITIVITY} \\
&= h_{\text{out}}\left(\sum_{i \notin \omega} h_{\text{inn}}^{i, \omega_i}(x_1^i) - h_{\text{inn}}^{i, \omega_i}(x_2^i)\right) && \text{SEE BELOW} \\
&= h_{\text{out}}\left(\sum_{i=1}^{d} h_{\text{inn}}^{i, \omega_i}(x_1^i) - h_{\text{inn}}^{i, \omega_i}(x_2^i)\right) \ . && \text{LET } h_{\text{inn}}^{i, \omega_i}(x) = 0 \text{ for all } i \in \omega
\end{aligned}
$$

Note that the function $h_{\text{inn}}$ may change from step to step above, the conclusion is only that such a decomposition exists. To see the step marked SEE BELOW, observe that the terms that condition on $\omega$ are capable of additively representing any function over $\omega$, so the explicit first term, i.e., that involving $h_{\text{inn}}^\omega$, may be omitted. Finally, observe that for the special case of noninteractivity, we have $\omega = \emptyset$, thus this first term always cancels out, and the above telescoping decomposition consists of exactly $d$ terms, one per feature. $\qquad\square$

We now show theorem 3.6.

**Theorem 3.6** (Axiomatic Models). *Suppose axioms 1–4. The following restrict $\mathcal{H}_{inn}$ or each $\mathcal{H}_{inn}^{(i)}$.*
*1. Suppose $\sigma$-linearity. Then $\mathcal{H}_{inn}^{(i)} = \{x \mapsto wx \,|\, w \in \mathbb{R}\}$, thus $\mathcal{H} = \{x_1, x_2 \mapsto \sigma(\boldsymbol{w} \cdot (x_1 - x_2)) \,|\, \boldsymbol{w} \in \mathbb{R}^d\}$.*
*2. Suppose* monotonicity. *Then $\mathcal{H}_{inn}^{(i)} = \{h : \mathbb{R} \to \mathbb{R} \,|\, x \leq y \implies h(x) \leq h(y)\}$.*
*3. **Multivariate Monotonic Models**: If we assume monotonicity but relax noninteractive compositionality, then $\mathcal{H} = \{x_1, x_2 \mapsto \sigma(h(x_1) - h(x_2)) \,|\, h : \mathcal{X} \to \mathbb{R} \text{ s.t. } \vec{x} \preceq \vec{y} \implies h(\vec{x}) \leq h(\vec{y})\}$.*
*4. **Conditional GAM Tree** If we relax NC to CIC, then $h_{inn}(\cdot)$ may be represented as a hybrid model of a decision tree / GAM model, starting with a tree over $\mathcal{X}_\omega$, where each leaf contains a GAM over $\mathcal{X}_{\backslash\omega}$. Formally, we have $\mathcal{H}_{inn} = \left\{\sum_{i=1}^{d-|\omega|} \left(T_i(x^\omega)\right)\left((x^{\backslash\omega})^{(i)}\right) \,\middle|\, T \in \mathbb{R}^{|\omega|} \to (\mathbb{R} \to \mathbb{R})^{d-|\omega|}\right\}$.*

*Proof.* We first show item 1. We then find that it is more straightforward to show item 3, and finally to conclude item 2 as a corollary.

We first show item 1. Recall that the linearity assumption requires that there exists some $\boldsymbol{w}$ such that for all $x_1, x_2$, it holds

$$\sigma^{-1} \circ H(x_1, x_2) = \boldsymbol{w} \cdot (x_1 - x_2) \ .$$

Consequently, the set of all feasible decision rules is parameterized by $\boldsymbol{w}$, in particular, applying $\sigma(\cdot)$ to both sides of the above, it holds

$$\mathcal{H} = \left\{x_1, x_2 \mapsto \sigma(\boldsymbol{w} \cdot (x_1 - x_2)) \,\middle|\, \boldsymbol{w} \in \mathbb{R}^d\right\} \ .$$

Finally, observe that this multivariate hypothesis class decomposes as

$$\mathcal{H} = \left\{ x_1, x_2 \mapsto \sigma \left( \sum_{i=1}^{d} h_{\text{inn}}^i(x_1^i) - h_{\text{inn}}^i(x_2^i) \right) \middle| h_{\text{inn}}^i(x) = \boldsymbol{w}_i x, \boldsymbol{w} \in \mathbb{R}^d \right\} \ .$$

From this, we conclude both the form of $\mathcal{H}$ and of each $\mathcal{H}_{\text{inn}}^{(i)}$.

NB $\sigma$ of $\sigma$-linearity and $\sigma$-transitivity must be identical over the domain (up to isomorphism), as $\sigma$-linearity and $\sigma$-transitivity would otherwise be mutually incompatible.

We now show item 3.

Recall that we assumed $\sigma$-transitivity (theorem 3.4 item 2). It thus holds that

$$\mathcal{H} = \left\{ x^{(1)}, x^{(2)} \mapsto \sigma \left( \sum_{i=1}^{d} h_{\text{inn}}^i(x_1^{(i)}) - h_{\text{inn}}^i(x_2^{(i)}) \right) \middle| h_{\text{inn}}^i \in \mathcal{H}_{\text{inn}}^{(i)} \right\} \ .$$

Monotonicity requires that if $x_1 \preceq x_2$, then

$$H(x_1, x_2) \leq \frac{1}{2} \ .$$

In our factoring, this is equivalent to

$$h_{\text{inn}}(x_1) \leq h_{\text{inn}}(x_2)$$

Consequently, the set of all such functions satisfying monotonicity is

$$\mathcal{H} = \{x_1, x_2 \mapsto \sigma(h(x_1) - h(x_2)) \,|\, h : \mathcal{X} \to \mathbb{R} \text{ s.t. } \vec{x} \preceq \vec{y} \implies h(\vec{x}) \leq h(\vec{y})\} \ .$$

NB: this does not specify $\sigma$, but due to the assumed factoring, $\sigma$ must be a symmetric continuous CDF with full support.

We now show item 2.

As we now restore NC, we now begin with the stronger factored form

$$\mathcal{H} = \left\{ x^{(1)}, x^{(2)} \mapsto \sigma \left( \sum_{i=1}^{d} h_{\text{inn}}^i(x_1^{(i)}) - h_{\text{inn}}^i(x_2^{(i)}) \right) \middle| h_{\text{inn}}^i \in \mathcal{H}_{\text{inn}}^{(i)} \right\} \ .$$

We then apply the same reasoning as in item 3, now applied to $x_1$, $x_2$ that differ in only dimension $i$, to conclude that the $i$th model factor obeys

$$\mathcal{H}_{\text{inn}}^{(i)} = \{h : \mathbb{R} \to \mathbb{R} \,|\, x \leq y \implies h(x) \leq h(y)\} \ .$$

This concludes the result. $\qquad\square$

We now show theorem 3.7.

**Theorem 3.7** (Proportionality of Choice). *Suppose a probabilistic decision rule over $n \geq 3$ items that obeys the* choice axiom. *Then $H(\cdot, \cdot)$ admits a $\sigma$-transitive factoring with logistic $\sigma(\cdot)$.*

*Proof.* Suppose items $A$, $B$, and $C \in \mathcal{X}$. Essentially, we use the telescoping product

$$\frac{\mathbb{P}(A \text{ best})}{\mathbb{P}(C \text{ best})} = \frac{\mathbb{P}(A \text{ best})}{\mathbb{P}(B \text{ best})} \frac{\mathbb{P}(B \text{ best})}{\mathbb{P}(C \text{ best})}$$

Substituting in the proportionality axiom:

$$\frac{1}{\sigma(\sigma^{-1}(\mathbb{P}(C > B)) + \sigma^{-1}(\mathbb{P}(B > A)))} - 1 = \left( \frac{1}{\mathbb{P}(B > A)} - 1 \right) \left( \frac{1}{\mathbb{P}(C > A)} - 1 \right)$$

Figure 4: An example pairwise comparison presented to participants of Study One of the Kidney Allocation dataset.

Rearranging the equation:

$$\sigma(\sigma^{-1}(\mathbb{P}(C > B)) + \sigma^{-1}(\mathbb{P}(B > A))) = \frac{1}{1 + \left(\frac{1}{\mathbb{P}(B>A)} - 1\right)\left(\frac{1}{\mathbb{P}(C>A)} - 1\right)}$$

$$= \frac{1}{1 + \exp \circ \ln\left(\left(\frac{1}{\mathbb{P}(B>A)} - 1\right)\left(\frac{1}{\mathbb{P}(C>A)} - 1\right)\right)}$$

$$= \text{logistic}\left(\ln\left(\frac{1}{\frac{1}{\mathbb{P}(B>A)} - 1}\right) + \ln\left(\frac{1}{\frac{1}{\mathbb{P}(C>A)} - 1}\right)\right)$$

$$= \text{logistic}\left(\text{logit}\left(\mathbb{P}(B > A)\right) + \text{logit}\left(\mathbb{P}(C > A)\right)\right) .$$

We thus conclude that $\sigma$ is the logistic function. $\square$

## B  EMPIRICAL ANALYSIS – OTHER DETAILS

### B.1  DATASET AND PREPROCESSING

**Kidney Allocation Real-World Datasets.** As mentioned earlier, the real-world dataset we used was collected by Boerstler et al. (2024) to test moral judgments *stability*. Across two studies, multiple participants were presented with several kidney allocation scenarios across 10 days (60 scenarios per day). In Study One, each kidney allocation scenario contained profiles of two kidney patients (Patient A and Patient B) described by four features: (a) number of dependents (0, 1, 2), (b) life decades gained from kidney transplant (1, 2, 3), (c) number of alcoholic drinks per day (0, 2, 4), and (d) number of crimes committed (0, 1, 2). In Study Two, each patient is described by five features: (a) number of elderly dependents (0, 1, 2), (b) life years gained from kidney transplant (1, 2, 3), (c) years waiting for the transplant (0, 2, 4), (d) weekly work hours post-transplant (0, 1, 2), and (e) obesity level (0, 1, 2, 3, 4). Participants were asked to decide who should get the kidney when only one was available and both patients were eligible. An example of one such pairwise comparison is presented in Figure 4.

Out of the 60 scenarios presented to each participant per day, several scenarios were repeated. This includes 6 scenarios that were repeated twice per day across all days to test for response stability and 2 scenarios that were used for attention checks. The remaining scenarios were randomly chosen. Since the repeated scenarios were non-random, we removed them from the dataset to ensure that the underlying data distribution for each participant corresponds to the uniform distribution over the input space. Sessions where participants failed attention checks were also removed from the dataset.

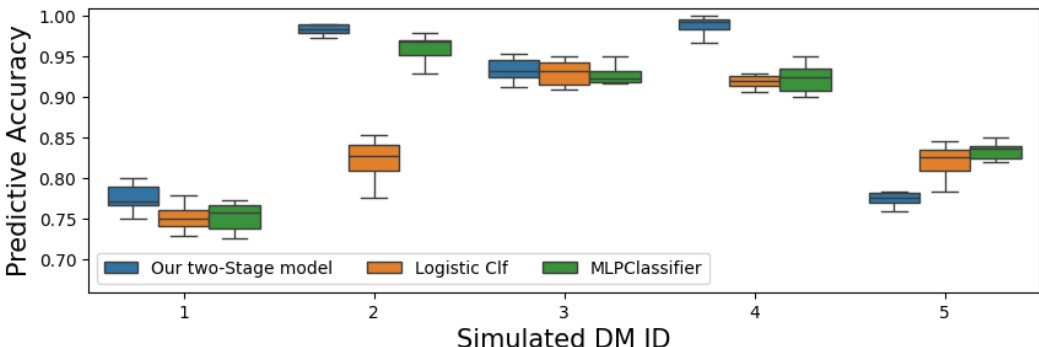

Figure 5: Comparison of our model vs baseline linear regression model for the synthetic dataset containing responses from five simulated decision-makers.

After the above steps, we had around 383 average responses from 15 participants in Study One and around 330 average responses from 40 participants in Study Two. This dataset is available under the CC-BY 4.0 license and is included with the Supplementary for the sake of completeness.

**Synthetic Datasets.** The synthetic dataset is constructed by simulating five decision-makers DM1–DM5, each using a different set of heuristics. The process for DM1 is described in Section 4. For DM2–DM5, the simulated decision processes are noted below. Recall that each decision-maker is presented with a pairwise comparison between two kidney patients, A and B, with features $x_A$ and $x_B$. Each patient is described using the patient's number of dependents, life years gained from the transplant, years waiting for the transplant, and their past number of crimes.

*DM2 simulation.* For DM2, the decision process for any pairwise comparison between A and B can be described as follows: (a) choose the patient with non-zero dependents; (b) if both patients have non-zero dependents, then choose the patient with greater value for life years gained; (c) if equal, then choose the patient who has been waiting longer for the transplant; (d) if equal, then choose randomly. Hence, this decision-maker also uses thresholds on the number of dependents and employs other features mainly for tie-breaking.

*DM3 simulation.* For DM3, the decision process for any pairwise comparison between A and B can be described as follows: (a) transform life years gained feature to reflect *diminishing returns*, i.e. $z_A^{\text{life gained}} = \lfloor \log\left(x_A^{\text{life gained}}\right)\rfloor$ and $z_B^{\text{life gained}} = \lfloor \log\left(x_B^{\text{life gained}}\right)\rfloor$; (b) assign $z_A^{\text{life gained}}$ points to patient A and $z_B^{\text{life gained}}$ to patient B; (c) assign $x_A^{\text{dependents}}$ points to patient A and $x_B^{\text{dependents}}$ points to patient B; (d) assign one point each patient who was been waiting for 5 years or more; (e) sum up the points for each patient and choose one with greater number of points (ties broken randomly). The log-transformation used in this decision-making process captures the diminishing returns property associated with the life years gained feature. Additionally, this process also uses the tallying heuristic to essentially count the number of factors favoring each patient.

*DM4 simulation.* For DM4, the decision process for any pairwise comparison between A and B can be described as follows: (a) transform life years gained feature to reflect *diminishing returns*, i.e. $z_A^{\text{life gained}} = \lfloor \log\left(x_A^{\text{life gained}}\right)\rfloor$ and $z_B^{\text{life gained}} = \lfloor \log\left(x_B^{\text{life gained}}\right)\rfloor$; (b) choose patient with greater $z^{\text{life gained}}$ value; (c) if equal, then choose the patient with more dependents; (d) if equal, then choose the patient who has been waiting longer for the transplant; (e) if equal, then choose randomly. This decision-maker also uses log-transformation for life years gained and other features for tie-breaking.

*DM5 simulation.* For DM5, the decision process for any pairwise comparison between A and B can be described as follows: (a) count how many features favor each patient and choose the patient favored by more features; (b) if equal, choose randomly. This decision-maker simply uses the tallying heuristic, choosing the option favored by more factors.

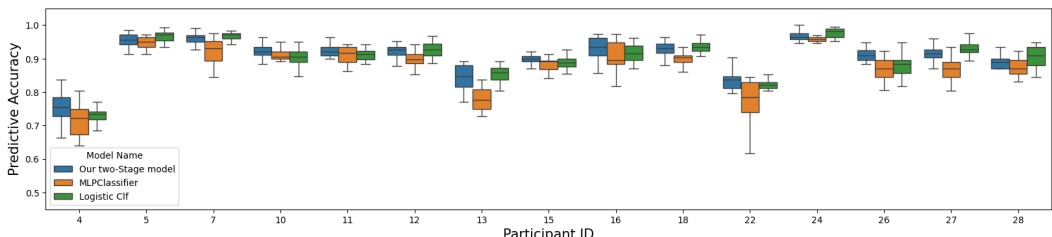

Figure 6: Comparison of our model vs baseline linear regression model for participants in Study One of the Kidney Allocation dataset.

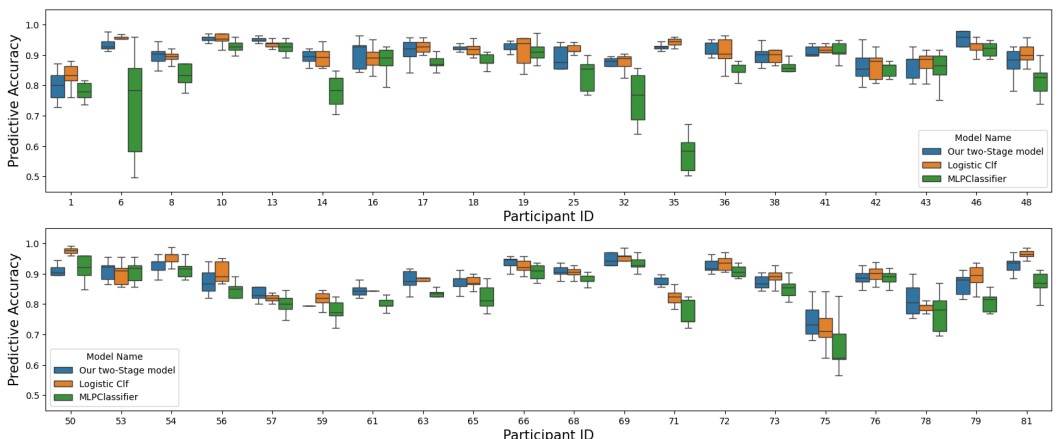

Figure 7: Comparison of our model vs baseline linear regression model for participants in Study Two of the Kidney Allocation dataset.

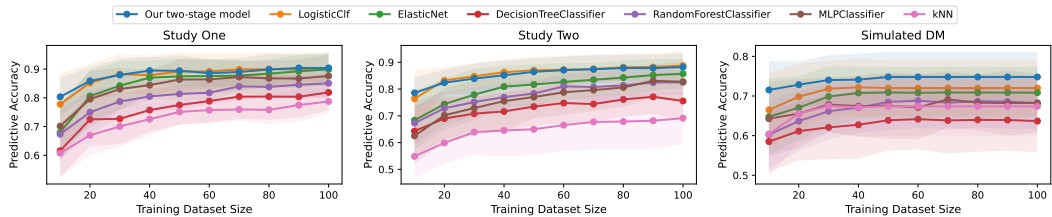

Figure 8: Comparison of our model vs baselines across increasing training size.

## B.2 IMPLEMENTATION DETAILS

For each decision-maker (real or simulated), we use a random 70-30 train-test split and report the summary statistics of predictive accuracy over test partitions across 20 repetitions.

**Our learning framework.** For our framework, we learn editing functions $h_{\text{inn}}^{i,\cdot}$ for feature $i$ that are designed to assign a score$\in \mathbb{R}$ to each value of feature $i$. As noted earlier, all $h_{\text{inn}}^{\cdot,\cdot}$ are constrained to be monotonic. Context $\omega$ is limited to one feature for the real-world dataset. The specific context is chosen using cross-validation. 20% of the training data is held out to use for validation purposes and the conditional two-stage model is learned for each setting of $\omega = \{i\}$ for all $i \in [d]$. The context $\omega = \{i^\star\}$ for which we achieve the smallest predictive error over the validation set is chosen as the final context for the corresponding participant. Context $\omega$ is kept empty for the synthetic dataset. For the dominance testing function $h_{\text{out}}(\cdot)$, we implement the simple tallying heuristic (i.e., the outputs from each feature-level editing function $h_{\text{inn}}^{\cdot,\cdot}$ are simply averaged). The final binary prediction is basically whether the $h_{\text{out}}(\cdot)$ is greater than 0 or not (with positive value favoring the patient on the left and negative value favoring the one on the right).

The two-stage model is trained by minimizing the chosen loss function (cross-entropy or hinge) with regularization and constraints. The regularization term quantifies the difference between editing functions learned for different values of the context variable $\omega$. We use this regularizer to ensure that editing functions for any feature $i$ corresponding to different context values are not too far from each other. As we noted earlier, $\omega$ is set to contain only one feature in our experiments on real-world datasets. For any two feature values $x^\omega = a$ and $x^\omega = b$, and for any other feature $i \notin \omega$, the regularization term measures $||h_{\text{inn}}^{i,a} - h_{\text{inn}}^{i,b}||$, for all $a, b$. The difference between the two editing functions is calculated by taking the squared norm of the difference between the outputs of these functions over all values in $\mathcal{X}_i$. Hence, the overall regularization term we use is

$$\lambda \cdot \sum_{i \notin \omega} \sum_{a,b} ||h_{\text{inn}}^{i,a} - h_{\text{inn}}^{i,b}||.$$

Here $\lambda$ is the regularization parameter and is set to be 1e-3 in our experiments. Additionally, we impose monotonicity constraints on each of $h_{\text{inn}}$ functions while optimizing the above loss; i.e., for all features $i$, $h_{\text{inn}}^i(a) \geq h_{\text{inn}}^i(b), \forall a > b \in \mathcal{X}_i$ OR $h_{\text{inn}}^i(a) \leq h_{\text{inn}}^i(b), \forall a > b \in \mathcal{X}_i$ (same constraints for context-based inner functions as well). We solve this constrained optimization problem using the Python SLSQP library (options 'ftol' and 'maxiter' are set to 1e-7 and 300, respectively).

**Baseline details.** The drift diffusion model was implemented using the Python PyDDM library with linear drift functions. The Bradley-Terry framework was implemented using the Python choix library, with a two-layer neural network scoring function.

We also implemented the following supervised modeling strategies to compare our method against. (a) Logistic Classifier – we implement the standard logistic classification approach, but impose a symmetry constraint by regressing over feature differences across the pairwise comparison; (b) Elastic-net Classifier – using the standard logistic classification approach over all given features with L1 and L2 regularization, with L1 ratio (scaling between L1 and L2 penalties) set to be 0.5. (c) Decision Tree Classifier – over all features with Gini splitting criterion; (d) Linear SVM – over all features with L2 penalty and squared hinge loss; (e) kNN – over all features with $n = 5$ neighbors; (f) Random Forest Classifier – over all features with 100 estimators; (g) MLP Classifier – over all features with two hidden layers of 10 nodes each.

**Computing resources used.** All experiments were run on a MacBook M2 system with a 16GB of random-access memory.

## B.3 ADDITIONAL EMPIRICAL RESULTS

**Performances and models for simulated decision-makers DM2–DM5**. As mentioned earlier, we also created a synthetic dataset containing responses from five simulated decision-makers. The descriptions of the simulated DMs are provided above. In this section, we report additional results for these simulations.

First, the performance of our model, Logistic Classifier, and MLP Classifier are presented in Figure 5. Note for all but DM5, our model has better predictive accuracy than baselines.

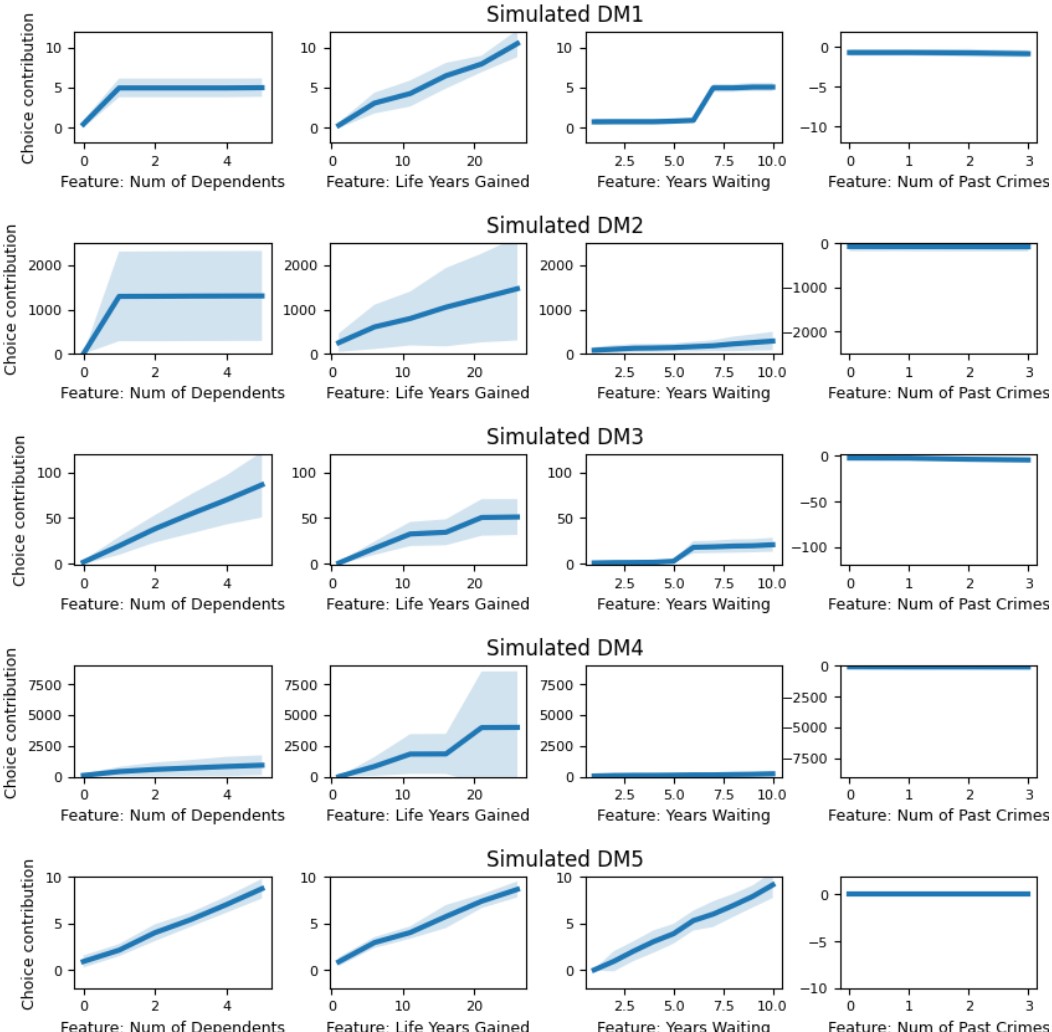

Figure 9: Editing rules learned by our approach for all simulated decision-makers DM1–DM5.

**Individual participant-level performances.** Across all participants, the performance of our two-stage model, Logistic Classifier, and MLP Classifier are reported in Figure 6 for Study One and in Figure 7 for Study Two (other baselines excluded here for presentation clarity). The plots show that our model has comparable accuracy to the Logistic Classifier for all participants across the real-world study datasets.

**Performance variation across training data sizes.** We also assess the performance of each model across variations of training size from 10 to 100. The results are reported in Figure 8. Our model reaches high accuracy levels faster than other models, with significant improvement in training performance compared to baselines for the simulated decision-makers.