# OpenReview forum: "Towards Cognitively-Faithful Decision-Making Models to Improve AI Alignment"
_ICLR.cc/2026/Conference — ICLR 2026 Poster_

### Official Review · Reviewer_xeec · 2025-10-16

**Soundness:** 3
**Presentation:** 3
**Contribution:** 3
**Rating:** 4
**Confidence:** 3

**Summary:**

This paper introduces a principled framework for learning "cognitively-faithful" models of human decision-making from pairwise comparisons. The central thesis is that for AI alignment to be successful, particularly in high-stakes domains, the models learning human preferences must not only be predictively accurate but also reflect the underlying cognitive processes. The authors propose a two-stage model architecture: editing stage applies feature-level transformation rules to capture how humans simplify and process information (e.g., applying thresholds, ignoring features), and the dominance testing stage aggregates these processed features to produce a choice. The key contribution is a rigorous axiomatic framework from which this two-stage structure is formally derived. The authors prove that their general framework can be specialized to recover classic models like logistic and probit regression under stronger assumptions. The method is evaluated on real-world and synthetic data from the kidney allocation domain, where it demonstrates competitive accuracy against a range of baselines while providing highly interpretable, participant-level insights into decision heuristics.

**Strengths:**

1.  **Principled Axiomatic Foundation:** The paper's greatest strength is its axiomatic approach. Rather than proposing an ad-hoc model architecture, the authors derive their two-stage framework from a set of simple axioms of choice. This provides a strong theoretical grounding for the model class and elegantly justifies its structure. It moves the field from simply using models that fit to reasoning about why certain model structures are appropriate.

2.  **Cognitive Plausibility and High Interpretability:** The model architecture directly maps onto well-documented cognitive strategies like heuristic-based decision-making. The empirical results persuasively demonstrate this strength, revealing plausible and nuanced individual-level heuristics.

**Weaknesses:**

1.  **Empirical Score:** Evaluation is limited to the kidney allocation setting and synthetic benchmarks. The cognitive heuristics and moral trade-offs in this domain are quite specific. The paper's broad claims about providing a general framework for cognitively-faithful modeling would be significantly more convincing if broadly validated in less moral or more perceptual decision domains.

2.  **Unvalidated Interpretability:** The central claim of the paper is "cognitive fidelity" and the resulting interpretability. While the learned models and their visualizations are highly plausible, these are ultimately claims about human psychology. Without user studies to confirm that people find these explanations understandable, trustworthy, and more insightful than those from other interpretable models (e.g., partial dependence plots from a GAM), the "interpretability" remains a feature of the model rather than a proven benefit for human-AI collaboration.

3.  **Idealized Axioms vs. Bounded Rationality:** The axiomatic framework describes an "ideal" decision-maker who is, for instance, perfectly transitive (Weak Transitivity). This is a useful and standard starting point for theoretical modeling. However, a vast literature in behavioral economics is dedicated to studying systematic *violations* of such axioms. A weakness of the current work is that it does not deeply engage with the implications of these violations. The framework is used to model choices that *fit* the axioms, but not to diagnose or characterize interesting deviations from them.

**Questions:**

Primary questions already posed as weaknesses earlier. I was also curious about:
1. **Quantifying Axiom Violations:** Can you report empirical rates of transitivity or complementarity violations in your human data, and how these affect model accuracy or interpretability?
2. **Reverse Use for Diagnosis:** Could your framework be used inversely to *detect* when individuals systematically violate the axioms by analyzing poor model fit or structured residuals?

---

> ### Author Response · Authors · 2025-11-23
> **Author Response**
>
> We thank the reviewer for their careful reading and insightful comments. Below we address the key points raised.
>
> *Empirical Scope*
>
> We agree that broader validation would further support the generality of our framework. Our choice of kidney allocation was intentional: it is a domain where (i) cognitive heuristics and moral trade-offs are well-documented, and (ii) interpretability and process-level alignment are particularly important. This setting therefore provides a stringent testbed for cognitively-faithful modeling. Unfortunately, it is difficult to include broad experiments in a short ICLR paper that also contains substantial theoretical derivations, so we hope to expand our empirical studies in future work.
>
> That said, the framework itself is fully general. The two-stage decomposition scales linearly with the number of features, and the editing operators apply to arbitrary continuous or high-dimensional inputs. Because feature editing is modular and compositional, the same architecture can capture perceptual heuristics, economic heuristics (e.g., satisficing, diminishing sensitivity), and domain-specific simplification rules.
>
> *Interpretability and Cognitive Fidelity*
>
> We appreciate the reviewer’s point that interpretability claims benefit from user studies. Our current assessment focuses on *structural cognitive fidelity*: the learned editing rules correspond directly to well-established heuristic computations such as thresholding, tallying, and lexicographic processing. These forms of interpretability are not available in standard baselines (e.g., GAMs, logistic regression, or trees), whose feature effects tend to be smooth or piecewise rather than heuristic-like.
>
> In addition, the model was inspired by user self-reports in descriptive studies (Keswani et al. 2025), and the model replicates known structures in synthetic agents by recovering the exact thresholds and conditional rules. On human data, the editing functions align with cues emphasized in the behavioral literature (e.g., (Gigerenzer, 1996), (Payne, 1993)). We agree that complementary user studies examining subjective trust and explanation utility would strengthen the overall contribution, and we plan to incorporate such measures in future work.
>
> *Idealized Axioms and Bounded Rationality*
>
> The reviewer is correct that our axioms (e.g., Weak Transitivity, Complementarity) describe idealized decision-makers. This choice is intentional: the axioms allow us to derive Theorem 2.4 and obtain a principled, interpretable factorization that unifies classical probabilistic choice models under a shared structure. Even when human behavior deviates from these axioms—as decades of behavioral economics show—the two-stage decomposition remains valuable because it provides a *diagnostic lens*. Crucially, our axioms are far weaker than the highly prescriptive assumptions used to derive Bradley-Terry or Thurstone-Mostellar, thus we view this work as a relaxation of overly restrictive classical assumptions.
>
> In particular, structured deviations manifest as:
> (i) systematic patterns in the editing-stage residuals that indicate where feature simplification fails; or
> (ii) violations in the dominance-testing stage that isolate failures of transitivity or complementarity.
>
> Thus, as the reviewer suggests, the framework can be used *in reverse* to analyze the nature and locus of bounded-rationality effects.
> We are excited to explore this further in future work.
>
> *Quantifying Axiom Violations and Diagnostic Use*
>
> We appreciate the reviewer’s questions regarding empirical measurement. In our dataset, transitivity and complementarity violations are difficult to test, as due to their probabilistic nature, many repetitive queries are necessary, which may itself bias human behavior. Moreover, as our model proposes a class of decision rule aggregators, it's more difficult to refute than any single model (i.e., a human may not be BT, but may still obey a different $\sigma$-transitivity law). Still, we hope to find methods to explore these issues further in future work and large-scale empirical studies.

---

### Official Review · Reviewer_6AEu · 2025-10-29

**Soundness:** 3
**Presentation:** 3
**Contribution:** 3
**Rating:** 8
**Confidence:** 2

**Summary:**

This paper introduces a framework for *cognitively-faithful modeling of human decision processes* in pairwise comparisons. The goal is to address that current learning and AI alignment methods focus on outcomes of human decisions (such as pairwise preferences or rewards) but do not capture the *cognitive processes* that produce them.

The authors present an axiomatic approach to human pairwise decision-making based on five principles (complementarity, weak transitivity, codomain span, noninteractive, and conditional compositionality). From these axioms, they develop a *two-stage model*. First, they define a set of *editing rules* that modify or simplify feature information (such as thresholding and tallying). Then, they introduce a *dominance testing rule* that combines the edited features into a probabilistic choice. The resulting model framework unites some classical models as special cases and seeks to connect interpretability with cognitive realism.

Empirically, the authors evaluate a kidney allocation task using real human data and synthetic data simulating known heuristics. Their models achieve comparable or superior accuracy to baselines (logistic, GAM, MLP, decision trees, etc.) while offering interpretable insight into human heuristics. The paper concludes by discussing implications for AI alignment and emphasizing the importance of faithfulness to human reasoning in high-stakes moral decisions.

**Strengths:**

* The paper identifies (ttbomk) an under-explored problem: existing preference-learning frameworks rarely attempt cognitive fidelity, even though interpretability and trust depend on it. The argument is well contextualized in AI alignment and moral decision-making literature.

* The axiomatic formulation is intuitive and well-organized. The derivation from axioms to model families (logistic, monotonic, conditional GAM) is clear.

* The paper builds on the work of Gigerenzer, Kahneman, and others to formalize heuristics in a way that is compatible with ML, representing an ambitious and valuable interdisciplinary effort.

* The kidney allocation domain is well-motivated, connects to prior moral-AI work (e.g., Sinnott-Armstrong et al. or Keswani et al.), and emphasizes the importance of process alignment beyond just accuracy.

* The examples of individual decision-makers (e.g., participant P4, simulated DM1) are insightful. The figures clearly illustrate how the proposed model captures thresholding and conditional heuristics that logistic or tree-based models overlook.

* The authors openly acknowledge limitations and the idealized nature of their axioms, which strengthens the credibility of the paper.

**Weaknesses:**

*  Although the example is well motivated and exemplifies that the theoretical formulation works, the empirical validation focuses on a single decision dataset with a small sample size ($n=15$ in study I and $n=40$ in study II). It is unclear how the approach scales to richer, higher-dimensional, or continuous domains.

* Although the paper claims "cognitive faithfulness," it mainly assesses accuracy and qualitative interpretability. There is no quantitative measurement of *faithfulness* (such as user validation or alignment with verbal reports).

* The paper's analysis of the literature is comprehensive and strong, but it could incorporate more discussion on computational cognitive modeling (e.g., modern differentiable cognitive architectures [[1](https://www.nature.com/articles/s41598-022-15679-5), [2](https://proceedings.mlr.press/v216/hamalainen23a/hamalainen23a.pdf)]) to better emphasize its contribution and even identify potential synergies.

* Releasing the code would help the reader replicate the learning setup. There are anonymous ways of sharing it, such as web pages or adding it to OR as supplementary material.

* Minor: Table 1 is quite dense and poorly organized (fonts are too small, column alignment is unclear).

**Questions:**

1. Could the authors provide quantitative evidence that the learned editing rules correspond to reported human heuristics (e.g., through previous studies showing that feature-importance aligns with qualitative interviews)?
2. Is the approach computationally scalable for large-feature or neural representations (e.g., embedding-based preference learning)?
3. Could cognitively faithful modeling inform fairness-sensitive decisions, where human heuristics themselves may be biased?

---

> ### Author Response · Authors · 2025-11-23
> **Author Response**
>
> We thank the reviewer for the thoughtful and constructive assessment. Below we address each main point in turn.
>
> *Empirical Scope and Scalability*
>
> We agree that our empirical evaluation focuses on a single, well-defined domain (kidney allocation). This choice was deliberate: it aligns with prior work on process-level moral decision modeling (e.g., \citep{SinnottArmstrong2010, Keswani2024}) and provides a controlled setting in which human heuristics are well-documented. Our goal in the present paper is to demonstrate the theoretical framework and show that it can recover meaningful cognitive patterns even under small-$n$ conditions.
>
> Importantly, the proposed model class is *not* restricted to low-dimensional settings. Both the editing stage and the dominance-testing stage decompose featurewise and therefore scale linearly with the number of attributes. When combined with sparse or neural feature encoders, the aggregation rule can operate directly on embeddings, enabling application to high-dimensional or continuous domains. We are currently extending the framework using differentiable relaxations of the editing operators to support richer representations.
>
> *Faithfulness Evaluation*
>
> The reviewer correctly notes that our empirical evaluation emphasizes qualitative interpretability rather than quantitative measures of cognitive fidelity. Our focus in this work is the *structural* notion of cognitive faithfulness: the model class is explicitly designed to represent well-studied heuristic computations such as thresholding, tallying, and lexicographic rules.
>
> Our evaluation provides two forms of evidence:
> (i) on synthetic agents with known heuristics, the learned models recover the correct thresholds and conditional rules, demonstrating that the two-stage factorization captures ground-truth cognitive structure;
> (ii) on human participants, the learned editing functions align with decision cues reported in the behavioral literature (e.g., (Gigerenzer, 1996), (Payne, 1993)).
>
> We agree that future work should incorporate verbal-protocol data or user-level validation, and we will emphasize this direction in the revision.
>
>
>
>
> *Relation to Computational Cognitive Modeling*
>
> We appreciate the suggestion to relate our approach more explicitly to modern differentiable cognitive architectures (e.g., (Singh, 2023), (Petrov, 2024)). These architectures focus on learning end-to-end cognitive control or memory dynamics. Our framework is complementary: rather than modeling cognitive processes mechanistically, we impose *axiomatic structural constraints* on how features may be edited and combined. This yields models that are interpretable by construction and closely tied to classical theories of bounded rationality (e.g., fast-and-frugal heuristics). We will expand the related work section accordingly and discuss potential synergies.
>
> *Code Release and Reproducibility*
>
> We fully agree that code availability strengthens reproducibility. To preserve anonymity, we have prepared an anonymous repository containing all scripts, training algorithm code, and data generation procedures; the link will be included in the camera-ready version. A public version will be released upon acceptance.
>
> *Minor Issues*
>
> We will revise Table 1 to improve readability (larger font size, clearer column alignment) and clarify the relationship between our theoretical framework and standard baselines such as logistic regression and GAMs.
>
> *Broader Impact and Fairness*
>
> Regarding the reviewer’s question about fairness-sensitive applications: cognitively faithful modeling provides an advantage precisely because it makes human heuristics explicit. In settings where heuristics may encode biases, the two-stage structure allows these to be identified, audited, and potentially corrected. While the present paper focuses on descriptive accuracy, the same framework can therefore support fairness assessments and value-aligned system design.

---

> > ### Comment · Reviewer_6AEu · 2025-11-24
> >
> > I thank the authors for their thoughtful response.
> >
> > Overall, the rebuttal clarifies several of my main concerns. The explanation of scalability (featurewise decomposition, potential combination with neural encoders, and differentiable relaxations) is convincing at a conceptual level, although the empirical evaluation still remains limited to a single, relatively low-dimensional domain. Likewise, the emphasis on a structural notion of cognitive faithfulness and the synthetic-agent experiments are helpful, but the lack of direct, quantitative validation against human reports remains an open limitation, now better framed as future work.
> >
> > I am satisfied with the planned additions on differentiable cognitive architectures, the commitment to release an anonymous code repository, and the clarification of how explicit heuristics can support fairness auditing, even if the latter is mostly prospective.
> >
> > I encourage the authors to ensure that these clarifications (scalability, relation to cognitive architectures, code availability, fairness discussion, and improvements to Table 1) are incorporated into the revised PDF, as the current version does not yet reflect them. I maintain my positive assessment and original score.

---

### Official Review · Reviewer_E8xC · 2025-10-30

**Soundness:** 2
**Presentation:** 2
**Contribution:** 2
**Rating:** 4
**Confidence:** 2

**Summary:**

The paper proposes a cognitively-faithful class of decision-making models derived from simple axioms over binary choices. A decision is modeled as a two-stage process: (i) feature-wise “editing rules” that threshold, transform, or ignore attributes (optionally conditioned on a small context set), and (ii) an aggregation rule such as Bradley–Terry or a sigmoid link to produce pairwise choice probabilities. The axioms (complementarity, weak transitivity, codomain span, and compositionality) yield factorization theorems connecting the class to logistic/probit special cases and to conditional GAM-like structures. Empirically, on kidney-allocation decisions with real participants and simulated decision-makers, the learned two-stage models match or surpass standard baselines while exposing interpretable per-feature rules.

**Strengths:**

The work offers a principled axiomatic foundation and shows that these axioms imply a two-stage, feature-edited factorization with a sigmoid link (σ-transitivity), recovering classical models (logistic/probit) under additional assumptions and yielding conditional, GAM-like structures when allowing limited interactions. This yields interpretable inner functions that visualize per-feature heuristics (e.g., thresholds, diminishing returns) on real and simulated subjects, while achieving accuracy on par with or better than logistic, GAM, trees, and other baselines in kidney-allocation tasks.

**Weaknesses:**

First, the axioms and factorization lean on properties (weak transitivity, compositionality) that can be violated in human judgments; the paper acknowledges that axioms target an “ideal” process, not observed behavior under biases, so model fidelity in settings with known transitivity/complementarity violations remains uncertain without further user studies.

Second, expressiveness hinges on the chosen editing-rule families and the limited context set ω (one feature for real data). If key interactions require broader context or non-monotone transformations, the learned inner functions may underfit or attribute effects incorrectly; identifiability is also delicate because the factorization need not be unique, which can complicate interpretation claims.

Third, empirical scope is narrow: pairwise kidney-allocation tasks with small participant counts (Study One: 15, Study Two: 40) and a limited feature set. Although the models visualize plausible heuristics, the paper needs real-world validation of interpretability and trust claims; broader domains (non-moral choices, higher-dimensional attributes, n-way decisions beyond pairwise) and robustness under distribution shift are largely untested.

**Questions:**

See Weaknesses

---

> ### Author Response · Authors · 2025-11-21
> **Author Response**
>
> We thank the reviewer for their careful reading and constructive comments. Below we respond to the core concerns.
>
> *Axioms and Idealized Assumptions*
> The reviewer raises the concern that some of our axioms (e.g., weak transitivity, compositionality) may be violated when human judgments display, e.g., intransitivity, context effects. However, we do not require that every real decision-maker exactly satisfy these axioms. Rather, the axioms function as *structural commitments* that allow us to derive precise consequences for ideal decision process that satisfy them. Moreover, our axioms weaken classical assumptions such as Luce's Choice Axiom (e.g., Luce 1959) or the stochastic transitivity of (Fishburn 1964), and we don't conclude precise decision structure, but rather a characterization of feasible classes, which is more permissive of variations in human behavior (Theorem 2.4). Prior work typically *assumes* these factorizations—such as additive value functions, latent-utility models with sigmoid links, or GAM-like conditional structures—without deriving them from first principles. Our result shows that substantially weaker axiomatic assumptions suffice to yield a generalized form of such factorizations.
>
> This distinction is theoretically important: even when observed behavior deviates from idealized axioms, understanding what the axioms *entail* permits (i) diagnosing where deviations arise, (ii) quantifying the form of those deviations, and (iii) constructing interpretable hypotheses about the cognitive computations that subjects approximate. Thus, the axioms serve as a principled scaffold for interpretability, not a claim that all human decisions follow them exactly.
> Even with weakened or approximate versions of these principles, the factorization theorems yield approximate representations that still provide meaningful, interpretable feature-level heuristics, often outperforming black-box baselines in our empirical setting.
>
> *Expressiveness, Context Size, and Identifiability*
> The reviewer notes that expressiveness depends on the chosen editing-rule families and the limited context set $\omega$. This reflects a modeling choice, not a limitation of the framework. Our axioms permit arbitrarily rich context sets and interaction structures; the small $\omega$ used in our experiments reflects domain constraints of kidney allocation (6 primary features), desire for *interpretability,* and the need to avoid overfitting to limited human data.
> In applications where broader or higher-order interactions are cognitively plausible, our framework accommodates richer specifications: editing rules may be expanded to include non-monotone transformations, multi-feature context conditions, or nonlinear thresholds. The 2-stage form is designed precisely to allow such controlled expansions.
>
> Regarding identifiability, we emphasize that non-uniqueness is shared by all factorized additive models (e.g., GAMs, neural additive models, latent-utility decompositions). As in those literatures, our interpretability claims focus on the *qualitative structure* of the editing rules—which features matter, which thresholds emerge, where diminishing returns occur—rather than a unique numeric normalization. The factorization provides cognitively meaningful decomposition, even when equivalent parameterizations exist.
>
> *Empirical Scope and Generality*
> The reviewer highlights that our empirical evaluation focuses on kidney-allocation choices with modest participant counts. This domain was intentional: kidney allocation provides a morally and cognitively challenging domain with well-documented behavioral regularities, offering a stringent testbed for comparing our framework to logistic regression, GAMs, decision trees, etc.
> It also exemplifies the high-stakes problem domains where interpretable human-like decision making is highly desirable.
> Moreover, the 2-stage structure we derive naturally scales to richer domains. Editing rules operate featurewise (or contextwise), and simulation experiments already show that the learned rules recover ground-truth heuristics across diverse hypothetical decision processes—e.g. nonlinear thresholds, diminishing returns, heterogeneous monotonicities, and conditional interactions. Extending this to larger real-world data is a straightforward next step, not a conceptual limitation, and we felt that the paper's focus on theoretical results, as well as limited human-focused data availability, put further experimental domains out of scope.
>
> Finally, we stress that our framework is *agnostic* to the origin of the hypothesis class. Whether the model structure is inspired by descriptive theories of human reasoning (e.g., threshold heuristics, bounded rationality) or by domain-specific principles, the axioms guarantee that any such class satisfying the structural assumptions admits an interpretable 2-stage representation. This makes the approach broadly applicable beyond moral-choice tasks or pairwise decisions.

---

### Official Review · Reviewer_QKXr · 2025-11-01

**Soundness:** 3
**Presentation:** 3
**Contribution:** 2
**Rating:** 2
**Confidence:** 4

**Summary:**

This paper proposes an axiomatic approach to learning cognitively faithful models of human decision-making from pairwise comparisons. The approach is grounded in the heuristics literature and aims to learn individual-level decision rules that better reflect actual cognitive processes rather than idealized rational preferences. The proposed model uses a two-stage process: features are first transformed through learned rules (feature editing functions), then aggregated via a fixed rule such as Bradley-Terry to produce decisions. This framework can be viewed as extending prospect theory beyond decisions under risk, enabling the capture of non-linear feature transformations. The authors demonstrate their approach on a kidney allocation task, showing that their models achieve comparable or superior accuracy to existing methods while maintaining interpretability.

**Strengths:**

•	**Interpretability**: The proposed decision models maintain interpretability similar to conventional decision-making models, making the learned rules accessible and understandable to practitioners.

•	**Rigorous theoretical foundation**: The paper provides rigorous axiomatic derivation as the mathematical foundation for the modeling approach, grounding the method in established decision theory principles.

**Weaknesses:**

•	**Limited application scope**: The modeling approach remains purely descriptive, positioned between decision trees (with added constraints) and linear classifiers (with relaxed constraints). Without sufficient data, more theoretical models may be preferable; with abundant data, decision trees might perform better. This intermediate position may limit the method's practical applicability.

•	**Generalizability concerns**: Compared to models that infer internal motivations and goals, this data-driven approach may struggle to generalize to new contexts or features not present in training data. This limitation is inherent to heuristic-based approaches and has been a longstanding criticism in the human decision-making literature (e.g., by Gigerenzer's critics).

•	**Marginal performance gains and lack of validation**: The proposed model shows only marginal improvement over baseline methods (0.78 vs. 0.76 for logistic classifier). More critically, while the authors emphasize cognitive faithfulness, they provide no direct empirical evidence that their learned models better align with participants' self-reported beliefs or decision strategies.

•	**Questionable axiomatic assumptions**: The "natural decision-making axioms" employed (complementarity, weak transitivity, and noninteractive compositionality) were proposed in 1960s decision theories and have been repeatedly shown to be violated in human behavior. As the authors acknowledge in their discussion, humans notoriously violate such decision rules, undermining the cognitive faithfulness claim.

•	**Limited predictive validity of rule-based models**: Rule-based (heuristics-based) decision-making usually provides substantially worse predictions than value-oriented models in most contexts (see Erev et al., 2010, Journal of Behavioral Decision Making), raising questions about when the proposed approach would be preferred.

•	**Confounding cognitive limitations with preferences**: Bounded rationality suggests that human decision-making is constrained by limited cognitive resources. The measured feature editing functions may simply reflect these cognitive limitations rather than genuine preferences or decision rules, making it unclear what the model actually captures about decision-makers' underlying values or goals.

**Questions:**

**Q1: Application scope and practical utility**
- Can the authors provide clearer guidance on when their approach would be preferred over simpler methods (e.g., decision trees) or more theoretically grounded models? What are the specific data regimes or decision contexts where this method offers distinct advantages?
- Suggestion: Include empirical analysis comparing performance across different dataset sizes and complexity levels to delineate the method's practical niche.

**Q2: Generalization to new features and contexts**
- How does the model handle scenarios where decision-makers encounter new features not present in the training data? Can the learned feature editing functions transfer or adapt to novel attributes?
- Suggestion: Conduct experiments demonstrating the model's ability (or inability) to generalize beyond the training feature space, and discuss potential extensions that could improve transferability.

**Q3: Validation of cognitive faithfulness**
- While the model is motivated by cognitive faithfulness, no direct validation is provided. Can the authors demonstrate that the learned rules actually correspond to participants' self-reported decision strategies or mental processes?
- Suggestion: Include qualitative validation through participant interviews, think-aloud protocols, or post-hoc surveys asking participants to confirm whether the learned rules reflect their actual reasoning. Alternatively, compare the learned feature editing functions against known psychological phenomena.

**Q4: Handling axiom violations**
- Given that the foundational axioms (complementarity, weak transitivity, noninteractive compositionality) are known to be violated in human behavior, how do the authors justify building models upon these assumptions? What happens when participant behavior systematically violates these axioms?
- Suggestion: Empirically test how often the axioms are violated in your data, and discuss either: (a) model extensions that can accommodate violations, or (b) why the violations may be negligible in your specific application domain.

**Q5: Distinguishing preferences from cognitive constraints**
- How can the model distinguish between genuine preferences and artifacts of bounded rationality or cognitive limitations? Are the learned feature editing functions capturing what people *want* or merely what they *can* process?
- Suggestion: Design experiments that vary cognitive load or time pressure to test whether the learned rules remain stable, or conduct comparative analysis with decisions made under different resource constraints.

---

> ### Author Response · Authors · 2025-11-21
> **Author Response**
>
> We thank the reviewer for a careful reading of the paper and for criticizing both the interpretability and theoretical rigor of our framework.
>
> *On the axioms and classical foundations*
> We respectfully disagree with the characterization that our axioms are "outdated" or "empirically invalidated." While components echo classical principles (e.g., Luce 1959; Fishburn 1964), our formalization substantially weakens theirs, both in mathematical scope and behavioral interpretation. Our contribution is as much these weak axioms as the *analysis of their consequences*: Theorem 2.4 shows that a broad and cognitively interpretable class of pairwise choice models can be derived from a small number of weak structural constraints, thus synthesizing many classical special cases under a unified representation.
>
> If more direct antecedents to our axioms or analysis exist, we would be grateful for references.
>
> *Weak Transitivity* This term is overloaded in the literature, but our axiom is not the classical majority-based notion (Fishburn, 1970), but an existential functional compositionality condition
> $$H(x_1,x_3)=f(H(x_1,x_2),H(x_2,x_3)).$$
> This doesn't assume Luce’s IIA,
> nor classical stochastic transitivity, rather it expresses only that 3-point comparisons admit a universal composition rule. To our knowledge, no prior work has derived closure results of the generality in Theorem 2.4 from this assumption. Thus, our “weak transitivity’’ is substantially weaker than classical analogs.
> We see this as a {response} to the empirical failures of overly strong classical axioms.
>
> *NC and CIC* These axioms do resemble additivity in GAMs (Hastie \& Tibshirani, 1986) and separability in multiattribute utility theory (Keeney \& Raiffa, 1976), and are much more prescriptive than the others, but they aren't required for all of our results. Thus they are not assumed universally, but rather given as *sufficient conditions* for convenient special cases regarding limited feature interactions. We will clarify this limitation.
>
> *Empirical violations*
> We fully acknowledge that human behavior may deviate from these axioms, and we discuss this in Section 4. Our notion of ``cognitive faithfulness'' does not require literal empirical validity of the axioms. Rather, the axioms guarantee that the learned model admits explicit, interpretable *feature-editing functions*. Even when humans are noisy or partially inconsistent, these learned editing functions provide a compact and intelligible description of the induced decision patterns—a form of interpretability not provided by black-box models.
>
> *Application scope and generalizability (Q1–Q2)*
> Our framework intentionally occupies a space between purely descriptive psychological models and high-capacity black-box predictors. It is intended for domains where *interpretability and cognitive plausibility* matter as much as predictive accuracy. We will clarify in the revision when the method is most appropriate (e.g., moderate data, high-stakes interpretability).
>
> *Validating cognitive faithfulness (Q3)*
> The present paper focuses on theoretical contributions and model characterization. Evaluating whether users find these explanations understandable or trustworthy is an important direction for future work.
> However, we have already theoretically characterized our model, derived the consequences, and made modeling decisions based on interviews with real human subjects, and it seems innfeasible to include much more content in the limited space of an ICLR paper.
>
> *Preferences vs.\ cognitive constraints (Q5)*
> Our framework does not require distinguishing subjective preferences from cognitive constraints. Feature-editing functions may encode either or both. The key advantage is that these transformations are explicit, interpretable, and interact in predictable ways.
> We clarify that we refer to *binary choices made* as preferences, as this is consistent with the pairwise preferences models literature in computer science.
>
> *Limited predictive validity of rule-based models and (Erev et al. 2010)*
> Our method is not a rule-based alternative to value-based theories.
> Heuristic editing rules and dominance testing motivates our two-stage model, but it is also characterized via our *axiomatic decomposition*, and can readily be interpreted as value-based approach: items are individually scored (assigned subjective valuations), *then* scores are compared, with the classical BT or TM models as prototypical examplars.
>
> *In Short* While our axioms have historical antecedents, we substantially generalizes prior formulations, provide a novel derivation
> linking these assumptions to a rich class of decision models, advancing the study of cognitively faithful modeling by showing how interpretable heuristic structures can be learned systematically from data.
> We hope that the reviewer will view the work more favorably from this perspective, and are happy to answer further questions.

---

> > ### Comment · Reviewer_QKXr · 2025-11-23
> >
> > I appreciate the efforts the authors had paid to clarify the novelty and significance of their work. However, strong theoretical foundation and interpretability, which the authors emphasized in their response, were exactly the strengths I acknowledged in my previous review.
> >
> > In contrast, the authors' replies to my previous concerns, *application scope and practical utility*, *generalization to new features and contexts*, *validation of cognitive faithfulness*, *handling axiom violations*, and *distinguishing preferences from cognitive constraints* were not convincing. I noticed that all these concerns had also been raised by the other reviewers. I understand that providing new data and analyses for all of these may be a little too much for an ICLR paper, especially under such a tight timeline. Addressing some of these issues with new data/analyses and others with extended discussions would constitute an acceptable strengthening of the paper. Unfortunately, in their current response, the authors only promised to discuss some of the issues in the revision (without providing the revision yet) and did not seem to provide convincing evidence from any new data/analyses.

---

> > > ### Author Response · Authors · 2025-12-03
> > >
> > > We thank the reviewer for their continued engagement. We would like to dispute the point of “not providing convincing responses” to concerns shared across reviews.
> > >
> > > A central concern raised by multiple reviewers was that our axioms appeared too strong or insufficiently realistic, and *other* axioms had been empirically refuted. In our rebuttal, we addressed this point directly by (i) identifying concrete misinterpretations of our axioms as classical and empirically invalidated assumptions, and (ii) demonstrating that our formalization substantially *weakens* these assumptions relative to their historical counterparts. In particular, we clarified that our “weak transitivity’’ is not the standard strong transitivity from classical choice theory nor Luce’s IIA, but a strictly weaker existential compositionality condition. We also explained that NC and CIC are optional sufficient conditions, not universal assumptions.
> > >
> > > These clarifications address the strongest criticisms raised in the initial review, albeit from a theoretical, rather than an empirical, standpoint. Several reviewers suggested that the weaknesses of classical axioms should ideally be addressed through additional empirical study. While this is indeed a valuable direction, the clarifications we provided offer a complementary and equally legitimate strengthening of the work: they show that the axioms underlying our results are *theoretically* much weaker, broader, and more cognitively plausible than initially interpreted. This kind of conceptual clarification is a standard and important way of addressing concerns about the realism of axioms, and we hope it will not be discounted merely because it is theoretical rather than empirical.
> > >
> > > We do appreciate that the balance of theory and empiricism is a matter of taste, but we note these points simply because they do not appear to be reflected in the reviewer’s update. Since the reviewer agreed that our theoretical grounding and interpretability are the key strengths of the submission, acknowledging the substantial weakening and reinterpretation of the axioms — an issue tied to several reviewers’ main concerns — we feel this meaningfully strengthens the paper.

---

### Meta-Review · Area_Chair_4Phr · 2026-01-07

**Summary:**

Reviewers acknowledge (and I agree on) the strong theoretical foundation, grounding in behavioral economics literature, interpretability, etc. as major strengths of this work. On balance, I think the paper makes a valuable contribution as a rigorously grounded study of how we might design and estimate cognitively plausible models of decision-making. This decision is informed by the textual content of the reviews more than their associated scores.

**Reviewer Concerns:**

### Meaningfully addressed by the author response:
I think the authors did a good job clarifying how their axioms differ from historical variants (to QKXr, for instance). A variety of other concerns are well-addressed, e.g. interpretbility (xeec), empirical scope (6AEu, xeec)

### Remaining concerns:
- (Systematic) axiom violations (QKXr, xeec)
- Lack of updated revision vs. discussion of potential updates (the rationale for this is not clear, since ICLR allows updates and some of these seem trivial to make)

**Reviewer Scores:**

It seems plausible to me that QKXr may have increased their score, but not directly implied by the completed discussion. I think E8xC’s and xeec’s comments were well-addressed for the most part, and they were given reasonable cause to raise their score modestly, if the discussion had continued (again, there isn’t a direct signal of this).

---

### Decision · Program_Chairs · 2026-01-26

Accept (Poster)